# Predictive design of crystallographic chiral separation

Rokas Elijošius [1], Emma King-Smith [2], Felix A. Faber[1], Louise Bernier[3], Simon Berritt [4], William P. Farrell [3], Xinjun Hou [5], Jacquelyn L. Klug-McLeod[4], Jason Mustakis [4], Neal W. Sach [3], Qingyi Yang [5], Roger M. Howard [4] ✉ & Alpha A. Lee [1] ✉

The efficient separation of chiral molecules is a fundamental challenge in the manufacture of pharmaceuticals and light-polarising materials. We developed an approach that combines machine learning with a physics-based representation to predict resolving agents for chiral molecules, using a transformer-based neural network. In retrospective tests, our approach reaches a four to six-fold improvement over the historical - trial and error based - hit rate. We further validate the model in a prospective experiment, where we use the model to design a resolution screen for six unseen racemates. We successfully resolved three of the six mixtures in a single round of experiments and obtained an overall 8-to-1 true positive to false negative ratio. Together with this study, we release a previously proprietary dataset of over 6000 resolution experiments, the largest diastereomeric salt crystallisation dataset to date. More broadly, our approach and open crystallisation data lay the foundation for accelerating and reducing the costs of chiral resolutions.

Chiral molecules play an important role in materials and pharmaceutical science. While the two mirror images, or enantiomers, of a chiral molecule have identical physical and chemical properties, differences arise in a chiral environment, e.g. when interacting with plane-polarised light. Incorporating chiral molecules into hybrid organic-inorganic materials enables the design of specialised chiral optoelectronics, from circularly polarised light detectors[1–3] and emitters[4,5] to spintronics[6,7]. In medicinal applications, the human body is a chiral environment, as the fundamental building blocks of biology are chiral L-amino acids. Therefore, mirror images of the same molecule can have different metabolic rates[8], different target affinities[9], and have unique toxicology profiles[10]. The latter point in particular has driven the pharmaceutical industry to focus on single-enantiomer formulations[11].

Advances in asymmetric synthesis[12–15] allow the direct synthesis of even highly complex natural products in an enantiopure manner. However, in many cases, due to the cost of chiral catalysts or the complexity of asymmetric synthesis, it is more practical to produce a racemic mixture. The mixture is then separated into enantiopure molecules via a chiral resolution process.

On the manufacturing scale, diastereomeric salt resolution is the resolution technique of choice, provided that the compound resolved has an acidic/basic functional group[16]. In an ideal diastereomeric salt resolution, an enantiopure resolving agent forms a less soluble diastereomeric salt with one enantiomer of the racemate while the other enantiomer remains in solution. The fundamental challenge in diastereomeric crystallisation is finding an appropriate resolving agent for a given racemate. Despite a history that spans more than 100 years, to date, predicting the optimal resolving agent for a given substrate is an unsolved problem[17,18].

We developed a physics-based machine learning model that predicts the probability of success for a given racemate-resolving agent pair (Fig. 1). Our model combines representations based on snapshots from molecular dynamics (MD) with a transformer-based neural

[1]Department of Physics, University of Cambridge, Cambridge, UK. [2]EaStCHEM School of Chemistry, University of Edinburgh, Edinburgh, UK. [3]Pfizer Research & Development, La Jolla, CA, USA. [4]Pfizer Research & Development, Groton, CT, USA. [5]Pfizer Research & Development, Cambridge, MA, USA. ✉e-mail: Roger.Howard@pfizer.com; aal44@cam.ac.uk

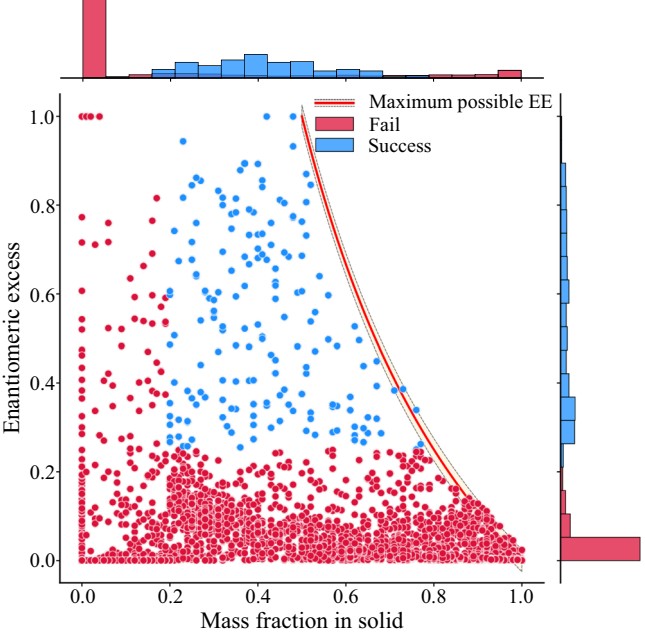

**Fig. 1 | Pair representations allow differentiation of enantiomers.** Our framework focuses on the representation of the acid-base pair, where differences between the two enantiomers naturally arise. A machine learning model uses the representations of both diastereomeric salts to predict the probability of success for a given resolution experiment.

network architecture. Compared to baseline methods, our model selects successful resolving conditions at a 4–6 times higher rate. We demonstrate that the interaction patterns identified by the network corroborate with X-ray crystal structures of the salt, indicating that the model has inferred physically salient interactions driving crystal formation. To prospectively validate our approach, we applied the model to identify promising resolving agents for six previously unseen racemates. In a single round of experiments, we identified successful resolution conditions for three of the racemates, while achieving an 8-to-1 ratio of true positives to false negatives among the full set of predictions tested.

## Results

### High-throughput data of diastereomeric resolutions across drug-like chemical space

As part of a larger push to improve the data landscape covered by high-throughput experimentation[19], we disclose more than 6000 previously unpublished chiral salt resolution reaction data acquired during ~10 years of medicinal chemistry synthesis support. The dataset includes 450 chiral compounds, which form more than 2000 unique acid-base pairs ranging from 12 to 72 heavy atoms in size (see Section S4 in the SI for an in-depth discussion of data diversity). The data also include a wide range of solvents; however, ethanol—the solvent of choice for initial screening—accounts for just over 40% of the data. To the best of our knowledge, previous data releases have been limited to specific molecules or scaffolds[20,21], as such, we believe our data constitute the largest diastereomeric salt crystallisation dataset released to date.

The results of an experiment are characterised by two figures of merit: the solid mass fraction (m.frac.) and enantiomeric excess (e.e.). The m.frac. corresponds to the fraction of the resolution substrate in the solid phase at the end of the resolution. The e.e. indicates the purity of the isolated solid and is defined as the difference between the molar fractions of the solid that incorporates one enantiomer versus the other.

$$\text{m.frac.} = \frac{m_{\text{solid}}}{m_{\text{initial}}}, \tag{1}$$

$$\text{e.e.} = |\chi_R - \chi_S|. \tag{2}$$

Here, $m_{\text{initial}}$ is the initial mass of the resolution substrate, and $m_{\text{solid}}$ is the mass of both enantiomers recovered from the precipitate. $\chi_R$ and $\chi_S$

**Fig. 2 | Low-mass-fraction and/or low-enantiomeric-excess data dominate the distribution.** The margins show the distributions of the mass fraction and enantiomeric excess for the successful and failed resolutions. Successful resolutions make up only 3% of the data. For clarity, we normalise the two distributions independently. The 'Maximum possible EE' line indicates the largest possible enantiomeric excess for a given solid mass fraction—points falling within 5% of the line—highlighted by the shaded area—were kept to allow for experimental error.

are the molar fractions of each enantiomer (($R$) and ($S$)) in the precipitate. The ideal resolution has both high m.frac. and e.e.—i.e. one and only one of the enantiomers is being fully crystallised.

However, in practice, the ideal resolution is seldom attained: Fig. 2 shows that most of the data lie in the low mass fraction, low enantiomeric excess region with 97% of the data having either m.frac. lower than 20% or e.e. lower than 25%, or both. A variety of factors can contribute to these outcomes. For instance, a low mass fraction could arise from slow crystallisation kinetics, where experiments are terminated before complete crystallisation, or from an inappropriate

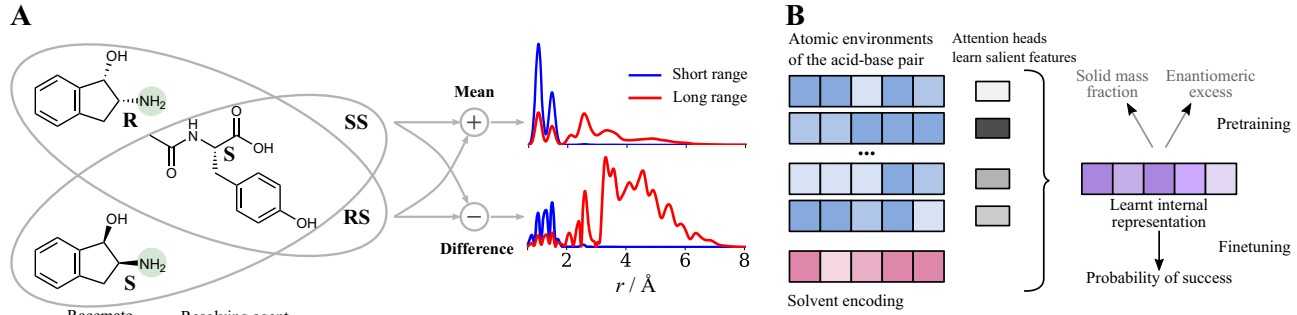

**Fig. 3 | Model architecture encodes chemical intuition. A** When interacting with a resolving agent, a racemic mixture forms two diastereomeric acid-base pairs. We create representations for both pairs and input their mean and difference into the model. The atom-density for the highlighted nitrogen shows that the mean of the representations focuses on the local environment around each atom, while the difference component emphasises the long-range intermolecular interactions. **B** The model uses transformer blocks to create an informative internal representation from the structures of the diastereomeric salts and the differences between them.

solvent choice, where both diastereomeric salts remain dissolved. Similarly, a low enantiomeric excess, even with sufficient solid material, can be attributed to phenomena like the formation of solid solutions or mixed salts. Given the high-throughput nature of our data collection, a detailed investigation of each experiment is not feasible. Therefore, for further modelling purposes, we operate under the assumption that all crystallisations have reached thermodynamic equilibrium and that the relative energy of the two diastereomeric salts is the primary driver of resolution success.

Unfortunately, this low figure of merit regime is also the regime where the data have high uncertainty (see Section S6 in the SI for details). High uncertainty in training data can introduce bias and significantly degrade model performance[22,23]. As such, we developed a specialised training procedure to combat this idiosyncratic source of noise.

## Chirality-sensitive machine learning

To predict the success of chiral separations, we focus on the diastereomeric acid-base pairs, rather than on the enantiomers and the resolving agents individually. Resolutions fundamentally depend on the energy differences between these diastereomeric pairs, which influence their relative lattice and solvation energies, and consequently, their relative solubility[24]. Of course, the relationship between diastereomeric salt pairs and their solubility is not always straightforward. Phenomena such as polymorphism and solvate formation mean that each diastereomeric salt pair can exist in multiple solid forms, each with potentially different energies[25]. Despite this added complexity, we hypothesise that by directly representing the acid-base pairs, our machine learning model can still effectively learn the key physical differences driving successful resolution, even without explicitly accounting for polymorphism and solvate effects.

A key challenge in representing acid-base pairs is capturing their vast conformational landscape. In addition to the conformations of the isolated molecules, we must consider the relative orientations between the two molecules. To address this, we generate MD trajectories for each unique enantiomer-resolving agent pair. Snapshots from these trajectories represent a diverse set of relative orientations, enabling us to incorporate dynamic information about the acid-base pair interactions.

To exploit the 3D information encoded in the MD trajectories, we employ an atom density representation[26–28]. Atom density representations encode the three-dimensional neighbourhood of each atom to yield a set of local atomic descriptors. These descriptors have been used extensively to directly predict molecular properties[29–33] or as a basis for other machine learning tasks[28,34–37]. Typically, these descriptors are most sensitive to each atom's closest neighbours. Here, we introduce dedicated long-range channels to capture intermolecular interactions in the acid-base pair, in addition to each atom's local environment. The final representation for a single atom consists of the mean and variance of its environment across the MD trajectory snapshots. This representation design emphasises the intermolecular interactions within the acid-base pair, aiming to provide a more direct and physically informed signal for downstream models.

Having obtained an informative representation, we train a neural network based on Transformer blocks[38,39] to predict the performance of diastereomeric resolution experiments. This choice of architecture is particularly pertinent given the nature of our acid-base pair representation and the underlying chemistry. A core aspect of our representation is its atom-centred nature, providing a set of local atomic environments as input. Transformer architectures are inherently well-suited to handle such set-like inputs. Furthermore, the success of diastereomeric resolution hinges on subtle, long-range intermolecular interactions within the crystal. The attention mechanism at the heart of Transformers is specifically designed to capture these complex, long-range dependencies, allowing the model to learn which intermolecular features are most predictive of resolution success. Beyond these architectural advantages for our specific problem, Transformers have also demonstrated broad applicability across diverse domains, from complex language tasks[40–44] to the natural sciences[45–50], further supporting our choice. In our case, the Transformer blocks act as trainable information filters, learning to prioritise which molecular environments/interactions drive crystal formation and which are less relevant. Specifically, the model uses representations of both diastereomeric salts formed in a resolution experiment (Fig. 3, Panel A), enabling it to consider both the overall structure and the key interaction differences that determine resolution success.

To address the uncertainty in the experimental data in the low figure of merit regime, we train an ensemble of models in two steps. First, we use all training data, including the subset with high uncertainty labels (i.e., solid mass fraction <20%), to train a regression model. Second, we fine-tune the last layer of each model using only the lower-noise subset of the training data, changing the training goal from regression to classification. This encourages the model to identify the acid-base pairs with the highest likelihood of success, rather than attempting to forecast noisy values. With this approach, the model first learns an informative internal representation and then leverages it to find the best resolution conditions. Panel B in Fig. 3 shows an abstracted version of the model's architecture and highlights the two-step approach.

Given the scarcity of resolutions with a truly high figure of merit in historical data (see Fig. 2), we consider data with m. frac. > 20% and e. e. > 25% to be successful. This partitioning still allows the model to prioritise fruitful resolution conditions while maintaining a better class

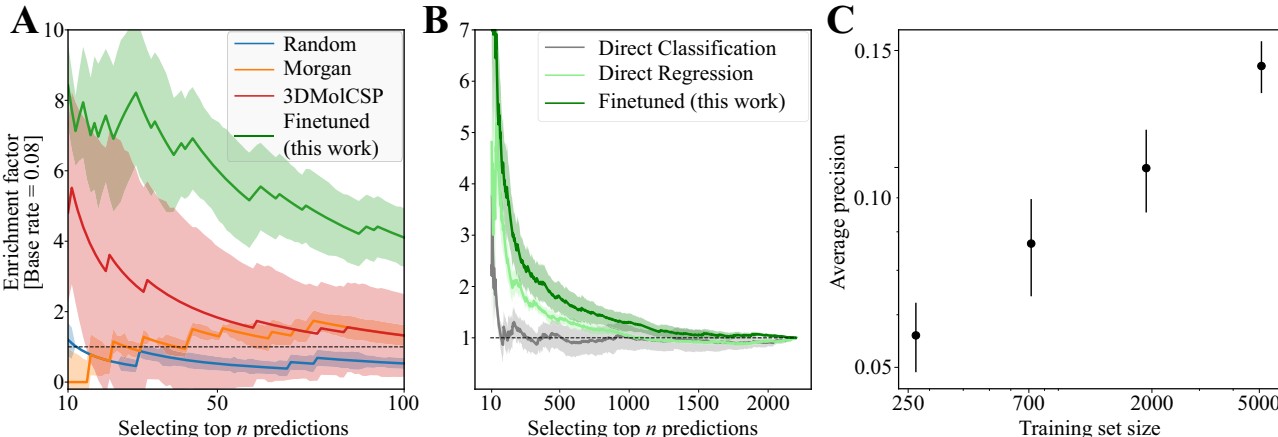

**Fig. 4 | Using the model quadruples the hit rate. A** Enrichment factor as a function of the number of selected top predictions, estimated using a 5-fold cross-validation experiment where each fold corresponds to unseen chiral salts. The blue line corresponds to encoding all of the participating molecules as random numbers, the orange line corresponds to encoding the molecular structures with Morgan fingerprints, the red line corresponds to another deep learning approach showing state-of-the-art results for chiral chromatography[51], and the green line corresponds to this work. The shaded area indicates the standard deviation estimated from an ensemble of five models. Note that the 8% base rate is the proportion of hits in the lower-noise subset of the data. **B** The model trained with the two-step approach outperforms the models directly trained on all training data, either as a classifier or a regressor. **C** Model performance, measured by average precision, systematically improves as more training data are added. Each point represents the mean performance of an ensemble of 10 models, and the error bars correspond to the standard deviation of the ensemble performance.

balance for training. See Section S3.3 in the Supplementary information for details on the construction of the ensemble.

## Model accurately predicts chiral separation for drug-like molecules

To evaluate our model, we use an enrichment factor metric, which quantifies how much better a model performs at finding hits compared to random guessing. Specifically, for a sample of size $N$, the enrichment factor is defined as the ratio between the number of hits in the model's top $n$ predictions and the average number of hits one would find by random sampling.

The test set for our model encompasses all the chiral mixtures present in the data through a modified 5-fold cross validation–ensuring each test fold contains only unseen racemates. Figure 4A shows that the model's top 100 predictions have an enrichment factor of four to six. The enrichment factor starts to decrease as we consider more predictions because most successful resolutions have already been found. By construction, the enrichment factor decays to one when all predictions are considered.

To contextualise our model's performance, we compare it against three baseline models of increasing complexity. We begin with a zero-knowledge Random Forest model, where each molecule is assigned a random barcode. This model serves as a control to address potential experimental design biases, which can be common in combinatorial datasets. Next, we consider a Random Forest model using Morgan fingerprints, which encode chemical motifs, including chirality (see Section S5.2 in the SI). This model acts as a sanity check to evaluate whether simple chemical patterns are sufficient to predict resolution conditions. Finally, we compare against 3DMolCSP[51], a deep neural network originally developed for chiral chromatography predictions. 3DMolCSP is the most complex baseline, as it can capture chirality, features a flexible neural network architecture, and has a slightly larger parameter count than our model, making it the most challenging comparison.

Figure 4A demonstrates that both neural network models outperform the simpler baselines, and importantly, our model surpasses 3DMolCSP. These results underscore the importance of chirality-sensitive representations and suggest the absence of data biases that could lead to trivial solutions. We hypothesise that our pair representation is a key factor in our model's superior performance compared to 3DMolCSP. Our representation can directly capture differences in the interactions of enantiomer pairs with the resolving agent, while 3DMolCSP must infer these effects from individual structures. Further details on 3DMolCSP modifications and model comparisons are available in SI Section S5. To ensure a fair comparison, we also adapted 3DMolCSP for diastereomeric resolutions and trained it using our two-step procedure.

Beyond the representational advantages highlighted in Panel A, a shared methodological choice contributing to the strong performance of both neural network models is the two-step training process. Panel B shows that the finetuned model outperforms both direct regression and direct classification. This training strategy is therefore crucial for achieving the results shown in Panel A and underscores the importance of a carefully designed training regime for complex prediction tasks like this.

Despite this effective training approach, the small number of successful resolutions in the training data remains a limiting factor for the model's performance; thus, a key question is whether the model is systematically improvable with more data. Figure 4C shows that the model's average precision improves as it is trained on more data (see Section S3.5 in the Supplementary Information for a more detailed discussion of the learning curve), and reassuringly without plateauing. Establishing a scaling relationship helps estimate the size of the training set required to achieve the target performance[52].

## Qualitative interpretation suggests the model captures relevant chemical contacts

Diastereomeric salt energy differences underpin successful resolutions. A key question is whether our model, trained solely on resolution outcomes, has learned physically meaningful interactions related to molecular packing that contribute to this energy difference.

Following established methods for interpreting attention mechanisms in molecular machine learning[53], we investigated correlations between the model's pairwise attention weights and the crystal structures of isolated chiral salts. We obtained X-ray crystal structures for four salts. Within each salt structure, we identified atom pairs within a 3.5 Å distance, defining these as 'neighbouring' atoms. We find that certain attention heads–components of the transformer blocks–

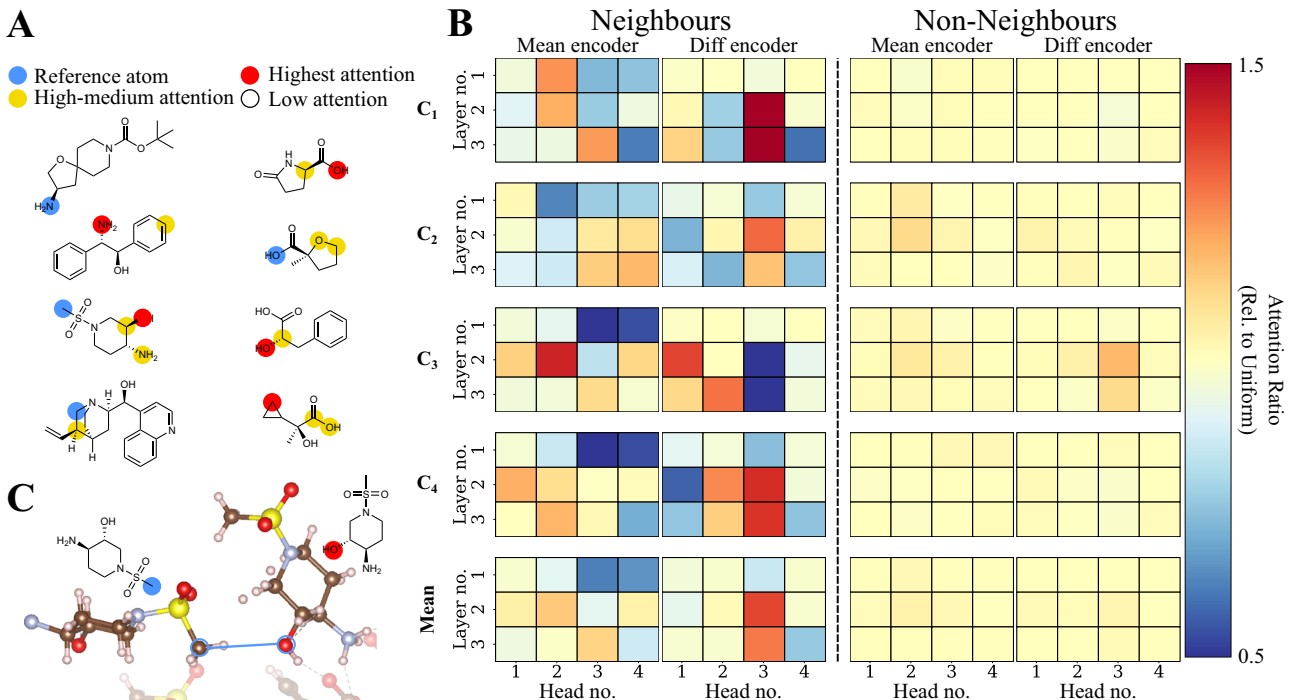

**Fig. 5 | Attention heads in the model recognise neighbouring atoms in the crystal. A** Examples of pairwise attentions in the model identifying close contacts in the crystal structure. Attention is calculated between the reference atom and the remaining atoms. In each example, we have confirmed that the reference atom and the highest attention atom are within 3.5 Å in the crystal structure. **B** The mean attentions between neighbouring pairs and non-neighbouring pairs for each attention head in the model across four different crystal structures. Some heads in the model focused on neighbouring pairs up to 50% more compared to the uniform baseline; meanwhile, no head focuses on non-neighbouring pairs. This result arises without the model ever seeing a crystal structure during training. We defined pairs as neighbours if they were within 3.5 Å in the crystal. **C** An excerpt from one of the X-ray structures, highlighting the non-trivial packing identified by the model.

focus on neighbouring atoms up to 50% more than the uniform-weights baseline.

In line with chemical intuition, one of the attention heads typically identifies the requisite carboxylic acid–amine interaction as illustrated by the top two examples in Fig. 5A. Intriguingly, other heads highlighted less obvious, yet crystallographically relevant, contacts. For instance, Fig. 5C demonstrates a case where the model assigns high pairwise attention between a methyl and a hydroxyl group, which ultimately end up adjacent in the crystal structure. Unlike the acid–amine interaction, there are no obvious intermolecular forces that push these groups together. In another example, one of the attention heads places a high weight between a carbon in the quinuclidine motif and a $CH_2$ group in the cyclopropyl ring of the acid. X-ray measurements confirm that the indicated methylenes are neighbours; however, they also show that a second cyclopropane carbon is a neighbour as well, which is an interaction the head did not capture.

Note that the specific attention patterns can vary between different initialisations (seeds) of the model. This variability suggests that within the high-dimensional solution space of our model, multiple sets of attention weights can lead to equally effective predictions, with different heads potentially learning to focus on different, but still relevant, patterns. Despite this variability, the general tendency for some attention heads to prioritise neighbouring atoms over a uniform baseline appears consistent across different model seeds.

The interpretability of attention weights remains a subject of ongoing debate[54,55]. However, we find it encouraging that our model—without explicit crystallographic training—qualitatively captures chemically meaningful spatial relationships. The observed tendency to assign higher attention to atom pairs that are close in the crystal structure provides a valuable sanity check, suggesting the model is sensitive to interactions relevant to molecular packing.

**Prospective model validation**

Having demonstrated the retrospective performance of the model and its connection to the underlying physics, we now turn to prospective validation. To test the model in a production-like scenario, we selected new unresolved chiral acids and bases and used the model to identify the resolving agents most likely to yield successful resolutions.

The final experimental design involved six diverse racemates, three acids and three bases, which were not present in the original data. Using the model, we performed the full combinatorial screen for the chosen molecules: each acid was screened against all chiral bases available to us as single enantiomers (and vice versa for the racemic bases) in all solvent systems present in the training data. The resulting virtual screen involved more than 20,000 resolution conditions. To select resolving agents for experimental validation, we ranked them according to the maximum predicted probability across all solvent systems and validated up to 13 best and worst resolving agents in four commonly used solvents (ethanol, acetonitrile, tetrahydrofuran, and ethyl acetate) for each racemate.

Nine conditions from the experimental screen resulted in successful resolutions. Eight of these used the model's top-ranked resolving agents, whereas one hit resulted from a bottom-ranked compound (Fig. 6A; Panel B shows the full confusion matrix). This suggests that using the model to guide the design of experiments is unlikely to overlook promising separating agents. Here, we impose a more stringent definition of successful resolution than during training to align with real-world requirements for optimisable conditions: success means that full dissolution is achieved before crystallisation takes place and that $z = \text{m.frac.}_{solid} \cdot \text{e.e.}_{solid} \geq 0.25$ ($z = 0.5$ corresponds to the ideal resolution), which occurs with only a 1% probability in the historical data (see Fig. S6 in the SI). As such, the eight hits found using the model's top-ranked acid-base pairs correspond to a three to 4-fold enrichment over the historic performance.

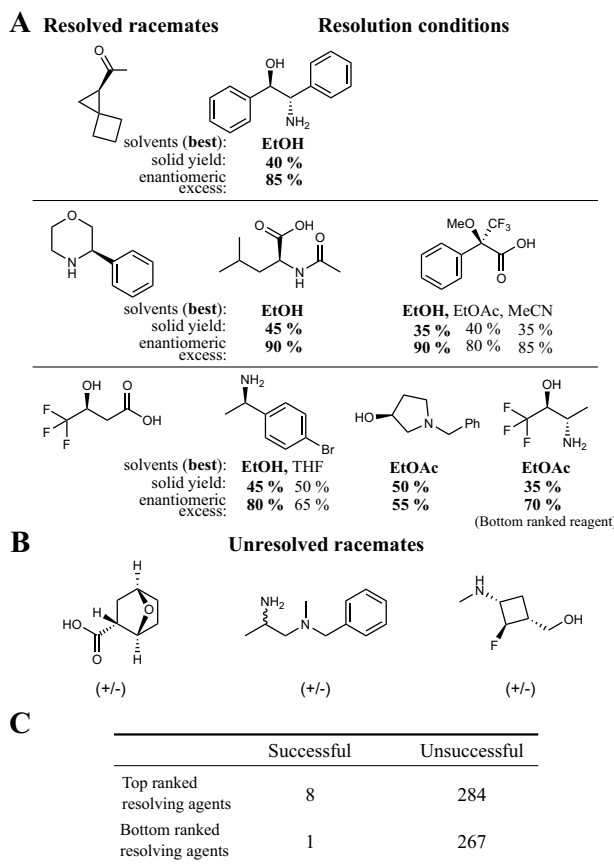

**A** Resolved racemates    Resolution conditions

solvents (**best**): **EtOH**
solid yield: **40 %**
enantiomeric excess: **85 %**

solvents (**best**): **EtOH**    **EtOH**, EtOAc, MeCN
solid yield: **45 %**    **35 %**  40 %  35 %
enantiomeric excess: **90 %**    **90 %**  80 %  85 %

solvents (**best**): **EtOH**, THF  50 %    **EtOAc**    **EtOAc**
solid yield: **45 %**  50 %    **50 %**    **35 %**
enantiomeric excess: **80 %**  65 %    **55 %**    **70 %**
(Bottom ranked reagent)

**B**    Unresolved racemates

(+/-)    (+/-)    (+/-)

**C**

|  | Successful | Unsuccessful |
|---|---|---|
| Top ranked resolving agents | 8 | 284 |
| Bottom ranked resolving agents | 1 | 267 |

**Fig. 6 | Model identifies successful resolutions in prospective experiments.**
**A** Structures of the resolved racemates and the conditions for the successful resolutions. **B** Unresolved substrates. **C** Full confusion matrix of the prospective experiment.

Notably, successful resolutions are not uniformly distributed among different racemates. In historical data, some racemates are resolved tens of times, while others are resolved only once or not at all (Section S4.2). We see a realisation of this in the prospective experiment, where three test molecules did not yield any successful resolutions. Crucially, our model demonstrates high recall, with all but one resolution discovered using the top-ranked resolving agents. This highlights its ability to effectively guide the identification of rare promising conditions in this needle-in-a-haystack problem.

The main limitation of our approach is the severe (but expected) lack of successful resolutions in the training data. As we show in Fig. 4C, the model's performance can be systematically improved with more data. Therefore, an iterative approach, in which the model is updated after each batch of new racemates and resolving agents is screened, could prove even more powerful.

## Discussion

We have developed a powerful method for predicting the outcomes of diastereomeric salt resolutions by combining physics-based representations with machine learning. Qualitative analysis of the model's attention weights provides encouraging evidence that it captures chemically relevant spatial relationships. Our approach is generally applicable and robust, as evidenced by both retrospective and prospective validation studies. More broadly, our work and the accompanying release of training data lay the foundation for a comprehensive solution to the long-standing challenge of predicting optimal resolving agents for diastereomeric salt resolutions. To further

enhance performance and address potential data limitations, future research could explore alternative machine learning paradigms such as multi-task learning or transfer learning approaches.

## Methods

### Experimental details

**Procedure for high-throughput experiments.** High-throughput experimentation reactions were set up inside an INERT Inc. triple double-sized glove box with $O_2$ and $H_2O$ levels <20 ppm. Glass vials (0.7 mL, 8 × 30 mm) pre-equipped with stir bars were used for each reaction. The reactions were set up with the components and conditions described by each dataset entry at 0.04 mmol scale and 0.2 M final concentration with a 1:1 ratio of acid to base and 200 μL total volume. The reaction vials were sealed by crimp under the glove-box environment and placed in a metal Chemglass Optichem 96-well heating plate atop a general IKA stirrer heater plate with an external temperature probe to accurately and evenly control the plate. The vials were heated for 1 h at 80 °C before being cooled over 3 h and left to stir for a further 15 h at 25 °C. Solubility observations were made after 1 h at 80 °C and after the 15-h stir period at 25 °C. At the reaction end point, the vials were centrifuged to settle any precipitates and the liquors were sampled and analysed by chiral SFC-MS for the determination of the liquor e.e. and the calculation of solids m.frac. and e.e. (individual methods detailed below, calculations aided by single-point calibration of racemate solutions of known concentration).

For any enriched liquor hits, the liquor was manually pipetted away from the centrifuged solid and the solid was re-slurried in fresh solvent (using the same solvent choice as the reaction, half volume, 100 μL) to rinse off residual liquor. The vial was centrifuged a second time, and the liquor (100 μL) was again removed from the centrifuged solid and combined with the original. Both the combined liquors and the residual solid were brought to an equal volume of 500 μL using MeOH to ensure solution before being analysed by chiral SFC-MS, with analysis versus a single-point calibration run in triplicate to reduce error. Measurement of the crystallised salt product % enantiomeric enrichment (ee) and % assay yield from rinsed and redissolved isolated solids superseded the product % ee and yield calculated based on SFC-MS analysis of uncrystallised material in the liquors.

iChem Explorer (Reaction Analytics, US) and Virscidian Analytical Studio™ software were used for data analysis.

HPLC-grade methanol, isopropanol, ammonium formate, and ammonia (Fisher Scientific, Pittsburgh, PA, USA), and bulk-grade carbon dioxide (AirGas West, Escondido, California, USA) were used in this study. The $CO_2$ was purified and pressurised to 1500 psig using a custom booster and purifier system from FLW, Inc. (Huntington Beach, CA, USA).

Analysis was performed using an Agilent 1260 SFC/MS system consisting of a binary pump, SFC control module, UV/DAD detector, and column compartment with an internal 6-position, 12-port valve, and 6120 MSD with an APCI source (Agilent, Inc., Santa Clara, CA, USA). A Gerstel MPS autosampler (Gerstel USA, MD, USA) was equipped with a 25 μL syringe and a 50 μL internal loop with a control method to vent $CO_2$ from the loop prior to sample introduction. The effluent of the SFC is split to the MSD using a 3-way tee (Valco, Houston, TX, USA) and a 50 cm long, 50 μm i.d. PEEKsil capillary tubing (Trajan Scientific, NC, USA) located between the column outlet and BPR. All data were acquired using Agilent 64-bit ChemStation (Version C.01.10).

**Characterisation of prospectively tested compounds.** The resolution experiments followed the general high-throughput experimental procedure described in the previous section. Characterisation details of the tested compounds are given in the Supplementary Information Section S7, including SFC-MS traces and X-ray deposition numbers when crystals were isolated.

## Molecular dynamics simulations of the acid-base pair

We initialise the simulation by placing the two molecules such that the distance between their closest two atoms is 2 Å. To ensure the two molecules remain within a few Ångstrom throughout the simulation, a restorative square potential is added. Before starting the sampling, we do an initial geometry relaxation, after which the simulation box is warmed to 500 K in 2.5 ps and a 0.25 fs time step. Afterwards, we sample conformations every 5 ps by running the simulation at a constant temperature regime with a 0.5 fs time step. Throughout the procedure, we use OpenMM[56] with the Sage forcefield[57] to calculate the forces and a Langevin thermostat to control the temperature. The molecules were kept neutral throughout the simulation (see Supplementary Information Section S1.1 for details).

For each racemate-resolving agent pair, we repeat the above procedure twice—once for each mirror image of the molecule that is being resolved. We continue the simulations until we have generated 200 snapshots for each pair. ZnTrack[58] was used to keep track of and organise both the MD runs and representation calculations.

Note, the models used for the prospective experiment originally used trajectories generated using PM7[59] with MOPAC[60]. We found that trajectories generated with the classical Sage forcefield resulted in close to identical representations while being substantially faster to generate.

## Representation details

**Atom-density representation.** Starting with a molecular dynamics snapshot, we transform the atomic coordinates into a set of local atomic environments. Each atomic environment is represented by a set $\{A_1(i), A_2(i), A_3(i)\}$, where $A_n(i)$ is an $n$-body atom density of atom $i$.

The $A_1$ term encodes the identity of an atom through nuclear charge, group, period, partial charge, and the population of $s$, $p$, and $d$ orbitals.

The $A_2$ and $A_3$ terms for atom $i$ are calculated by placing Gaussian functions on all neighbouring atoms within an 8 Å cutoff. Each Gaussian is multiplied by a weight that depends on the properties of the central and neighbouring atoms, such as the MOPAC-calculated $\sigma$ orbital overlap between the two atoms.

Equations (3) and (4) give the definitions of the $A_2$ and $A_3$ terms.

$$A_2(i)^\omega(r) = \sum_{j \neq i} \omega_{ij} G_{ij}(r),   \tag{3}$$

$$A_3(i)^{\omega, \phi}(r) = \sum_{j, k \neq i} \omega_{ij} \omega_{ik} \omega_{jk} G_{ij}(r) \phi(\theta_{ijk})   \tag{4}$$

Where $G_{ij}$ is the Gaussian function placed at atom $j$:

$$G_{ij}(r) = \frac{1}{\sigma \sqrt{2\pi}} \exp\left(-\frac{(d_{ij} - r)^2}{2\sigma^2}\right)$$

$d_{ij}$ is the interatomic distance of atoms $i$ and $j$, $\sigma$ is 0.0825 Å. $\omega_{ij}$ is a weighting function (explained in more detail below) and $\phi(\theta_{ijk})$ is the sine or cosine of the principal angle between atoms $i$, $j$, $k$—providing three-body information for the $A_3$ term.

The above expansion captures the spatial information around each atom even without the $\omega_{ij}$ weighting terms. The $\omega_{ij}$ terms further enhance this by including electronic information. The following weighting functions are used:

$$\mathcal{W} = \{\varepsilon^\sigma, \varepsilon^\pi, \varepsilon^\delta, q, p^\sigma, p^\pi, p^\delta\}$$

$\varepsilon_{ij}^*$ corresponds to the * orbital overlap between atoms $i$ and $j$. The $q$ term captures the electrostatic interaction as $q_{ij} = q_i \cdot q_j$. $p_{ij}^*$ is similar to the overlap term, but uses electron *populations* in * type orbitals in the two atoms rather than the overlap. These weights are extracted by running a MOPAC job with:

<pre>PM7 1SCF MMOK VECTORS PI BONDS DENSITY</pre>

The overlap terms $\varepsilon_{ij}^*$ are short-ranged as the orbital overlap decays exponentially as $d_{ij}$ increases. On the other hand, the $q$ and $p_{ij}^*$ terms do not depend on distance, thus capturing more long-range information. Together with the spatial information from the Gaussian function, this set of weighting functions provides a comprehensive representation of the spatial-electronic environment of each atom.

The full $A_2$ and $A_3$ terms correspond to the sets.

$$A_2(i) = \{A_2(i)^{\varepsilon^\sigma}, A_2(i)^{\varepsilon^\pi}, A_2(i)^{\varepsilon^\delta}, A_2(i)^q, A_2(i)^{p^\sigma}, A_2(i)^{p^\pi}, A_2(i)^{p^\delta}\},$$

$$A_3(i) = \Big\{ A_3(i)^{\varepsilon^\sigma, \cos}, A_3(i)^{\varepsilon^\pi, \cos}, A_3(i)^{\varepsilon^\delta, \cos}, A_3(i)^{q, \cos}, A_3(i)^{p^\sigma, \cos}, A_3(i)^{p^\pi, \cos},$$
$$A_3(i)^{p^\delta, \cos}, A_3(i)^{\varepsilon^\sigma, \sin}, A_3(i)^{\varepsilon^\pi, \sin}, A_3(i)^{\varepsilon^\delta, \sin}, A_3(i)^{q, \sin}, A_3(i)^{p^\sigma, \sin},$$
$$A_3(i)^{p^\pi, \sin}, A_3(i)^{p^\delta, \sin} \Big\}.$$

Where the individual terms are obtained by evaluating equation (3) or (4), respectively over 160 bins with $r$ ranging from 0.7 to 8 Å.

Repeating the process for every non-hydrogen atom in the snapshot furnishes the full representation for a single snapshot. The final representation is obtained by concatenating the mean and standard deviation of the individual snapshot representations.

**Solvent representation.** To represent the solvent, we use COSMO*therm*[61–64] calculated descriptors. These descriptors include a range of physical properties and the surface charge profile of the solvent (see Table S1 in the Supplementary Information for a complete list of descriptors). The surface charge profile describes the probability for a charge with a certain magnitude to appear on the surface of a molecule. Such profiles have been utilised for applications like asymmetric catalysis optimisation[65], adsorption/desorption modelling[66], and reaction kinetics studies[67].

Some resolutions involve solvent mixtures rather than pure solvents. While COSMOtherm can handle mixtures, separate calculations would be needed for each new combination. Since the solvent plays a secondary role to the enantiomer-resolving agent pair, mixtures are represented by a molar fraction weighted sum of the pure solvent descriptors.

## Model details

The model is designed to learn from the structural differences and similarities between diastereomeric acid-base pairs in various solvents. The overall architecture is depicted in Fig. S1 in the Supplementary Information.

The model's inputs are atom-density representations of the two diastereomers. As these initial representations are high-dimensional ($\approx 7000$ dimensions), we first compress them to a 20-dimensional vector using a pre-trained auto-encoder. From these compressed representations, the model constructs two key components: the mean of the two diastereomer representations and their absolute difference, as described in section 'Chirality-sensitive machine learning'. These components are independently processed through a series of transformer blocks before being aggregated via cross-attention using the solvent representation.

We employ a two-stage training procedure. The model is first pre-trained on the full dataset, after which it is fine-tuned on a high-confidence, low-noise subset. During this fine-tuning stage, all model weights are frozen except for the final feedforward network, allowing the model to adapt its readout without disrupting the learned internal

representations. To ensure robust evaluation, we use a 5-fold cross-validation scheme stratified by unique racemates. Given the significant class imbalance in our data (3% positive examples), we use weighted sampling to construct training batches with an equal proportion of successful and unsuccessful resolutions, which substantially improved training stability.

Hyperparameters were systematically optimised using a Tree-structured Parzen Estimator approach as implemented in Optuna[68]. A comprehensive list of the final hyperparameters, a detailed description of the training procedure, and the data splitting scheme can be found in Section S3 of the Supplementary Information.

## Data availability
The training data, related models and predictions for the prospective experiment are available at https://doi.org/10.5061/dryad.d2547d89c[69]. The deposition numbers of crystallographic data generated in this study are available in the Supporting Information. The Source Data file contains predictions for the retrospective tests and other data needed to reproduce the Figures. Source data are provided with this paper.

## Code availability
Code related to the manuscript can be found at: https://github.com/RokasEl/DiastereomericResolutions[70] under an MIT licence. Code for the modified 3DMolCSP model can be found at: https://github.com/RokasEl/3dmolcsp-diastereomeric-resolutions[71] under a CC BY-NC-SA 4.0 license.

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

## Acknowledgements

We thank W.McCorkindale and J.Riebesell for helpful discussions. R.E. acknowledges support by the University of Cambridge Harding Distinguished Postgraduate Scholars Programme and the Winton Programme for the Physics of Sustainability. Financial support for this work was generously provided by Pfizer and the Royal Society (Newton International Fellowship to E.K.S. and University Research Fellowship to A.A.L.) Access to computational resources was in part obtained through a University of Cambridge EPSRC Core Equipment Award EP/X034712/1.

## Author contributions

R.E. and F.A.F. carried out the computational work. R.E., F.A.F., E.K.S., and A.A.L. planned the computational studies. S.B., X.H., R.M.H., J.M., N.W.S., and Q.Y. conceptualised the data. L.B., N.W.S., and W.P.F. carried out the experimentation for the historical data and prospective experiment. R.M.H., N.W.S., Q.Y., and J.L.K.M. curated the data. R.E. and E.K.S. wrote the manuscript. A.A.L. and R.M.H. edited the manuscript. All authors reviewed the manuscript and contributed to the planning of the overall project.

## Competing interests

A.A.L. is a co-founder and owns equity in PostEra Inc. and Byterat Ltd. L.B., S.B., X.H., J.L.K.M., J.M., N.W.S., Q.Y., and R.M.H. are employed by Pfizer Inc. WPF is employed by Virscidian Inc. and formerly at Pfizer. F.A.F. is employed by AstraZeneca at the time of publication; however,

none of the work presented in this manuscript was conducted at or influenced by this affiliation. The remaining authors declare no competing interests.
