## [Transparent Peer Review file · Nature Communications]

Predictive design of crystallographic chiral separation

Corresponding Author: Dr Alpha Lee

Version 0:

Reviewer comments:

Reviewer #1

(Remarks to the Author)

This paper by Lee and coworkers focuses on the use of molecular dynamics and training a neural network to aid in the prediction of diastereomeric salt formation for chiral resolutions. A previously unreleased historical dataset of 6,000 resolution experiments forms the training set and demonstrates the difficulty in current screening for diastereomeric salt resolutions. Even with setting a very generous bar for "success" (>20% solid mass fraction and >25% ee), only 3% of the historical dataset falls in this category; highlighting the difficulty in the classical screening approach. Once built, Lee and coworkers use their model to resolve 3 racemates with higher success rate than the original dataset without the use of the model.

This predictive approach will be of interest to those seeking to access enantioenriched materials. Particularly since diastereomeric resolutions are still commonplace within the pharmaceutical industry. However, key points need to be addressed before publication of this work.

1. Pg1, Line 37 – Replace "acidic/basic centre" with "acidic/basic functional group"
2. Pg1, Line 38 – Replace "insoluble salt" with "less soluble diastereomeric salt"
3. I was very interested in the 6,000 resolution experiment dataset that was reportedly used in this work, however, I cannot find this dataset within the SI or the "additional data" zip file. I believe it would be beneficial to make this dataset visible in the SI, perhaps as a table, so that chemists can access it
4. Figure 2 – the 5% error line for the "maximum possible EE" is only visible when figure is magnified – perhaps the use of higher contrast colour or a black dotted line would make it more visible to the reader.
5. Did the neural network use any crystallographic data during training?
6. Figure 5C shows an X-Ray crystal structure. Has this data been collected by the authors? If it is an original crystal structure reported as part of the paper, the CIF needs to be deposited on the CCDC and the deposition number should be supplied for reviewing. If it is from a previously published structure, it requires a reference.
7. Pg 5, Line 151-152 mentions 3 more crystal structures. These need to be described within the SI and deposited into the CCDC. The deposition numbers are required for reviewing the obtained data.
8. Figure 6 switches to using %ee for reporting the enantioenrichment for the resolutions – perhaps it would be worth using % enantiomeric excess label in the figure.
9. Figure 6C – I'm not sure what new information this chart is conveying. The "needle-in-the-haystack" problem was already shown in Figure 2. The focus should be on demonstrating the resolutions predicted by the model instead of the historical dataset.
10. Pg 7, Line 171 mentions 6 racemates tested – 3 acids and 3 bases. How did the other acid and 2 bases perform that are not shown in Figure 6?
11. How was the configurational assignment of the resolved materials performed? Did the model predict the correct enantiomer?
12. Upon checking the SI, there are no obtained characterization data for the materials obtained after the resolution so there is no verification of the identity of any of the claimed compounds - No NMR spectra, chiral HPLC chromatograms, MS, IR, melting points, optical activity (optical rotation or specific rotation) or crystal structures are reported to prove the identity and purity of the resolution experiments. At a minimum the paper will require chiral HPLC (to prove ee) and ¹H NMR spectra (to prove identity and purity) and HRMS or crystal structures (to prove elemental composition). Currently, there is no experimental data in the manuscript or SI to show that the resolutions were successful.
13. The resolution experiments are conducted on a very small scale (0.04 mmol scale). For example, the 2-phenylmorpholine resolution with acetyl leucine, with a 45% yield, it is expected to yield ~6 mg of material. On this small scale (and without ¹H NMR spectra to verify) it is possible to get unrepresentative yields due to the presence of trace solvents. It would be preferable to perform these resolutions on a 0.5-1.0 mmol scale, to get more representative yields and

supply characterization data to verify the solids are dry when calculating yields from their obtained masses (or use an appropriate assay method, such as NMR assay with an internal standard).

As such, these points require addressing before publication. Most importantly, key experimental data has not been collected (characterization of compounds) or made available for the review process (no chiral HPLC traces of resolved materials, X-ray structure CIFs/CCDC deposition numbers, masses of obtained materials).

(Remarks on code availability)

Reviewer #2

(Remarks to the Author)

Manuscript title: Predictive design of crystallographic chiral separation

Manuscript ID: NCOMMS-24-52429-T

Recommendation: minor review

The authors have developed a physics-informed machine learning model to predict resolving agents for the diastereomeric classical resolution of chiral molecules, utilizing a proprietary dataset of 6,000 prior experimental resolution attempts. Although, as acknowledged by the authors, the dataset is biased, with significantly more 'failed' than 'successful' attempts, the modelling methodology is rigorous and carefully designed, and the paper is exceptionally well-written. I only have a few minor comments for consideration (not in order of importance):

Abstract line 15 and pg. 2 line 46: 4-6 times more accurate than what? The benchmark only becomes clear much later in the text, please consider adding something here for the abstract readers.

Figure 2: The graph shows that 97% of the test cases ended up being unsuccessful. In my view, a key question is also why is this the case? While indeed the solubility differences of the diastereomeric salt pairs drive effective separation (main scope of this work), there are other reasons why a resolution could fail. For example, low ee could mean that the diastereomeric salts never formed (e.g. due to the resolving agent being only weakly basic or acidic). Other possibilities are diastereomeric double salts, solid solutions, or stability issues. I know all this is probably beyond the scope of this work, but it would be nice to get a feeling for these effects too, and more details on the training dataset could help there.

Pg. 7, line 175 and data in Figure 2: Linked to the previous comment. Could you please add (probably in the SI) the list of chiral resolving agents, substrates (if possible/ if not at least total number of substrates involved, their MWs and/or Pka) and conditions used in the virtual and experimental screenings? From the experimental screenings, do you see any obvious patterns in the use of certain resolving agents vs success of resolution?

Figure 2: In classical resolution, it is often found that often < 1 equiv (e.g. 0.5-0.7 equiv) of resolving agent yields the most effective resolution conditions. If the racemate is well soluble, the less soluble diastereomeric salt will preferentially crystallize consuming the resolving agent and preventing the formation of the other diastereomeric salt. With this in mind, what conditions did the resolution experiments in fig 2 use in terms of resolving agent equiv? Also, is this effect something that the model could in theory capture?

Pg. 3 line 80: Relative solubility of the diastereomeric salt pairs will also depend on the occurrence of polymorphism and maybe even more pronounced the occurrence of solvates, both affecting the lattice and solvation energies. Thus, while looking at the different resolving agents is indeed the first step, solvent effects should not be neglected – which is often the case in such screenings. I understand the presented model cannot capture such effects, but something worth mentioning I believe.

Figure 4A: It seems a bit surprising that the “zero knowledge” model is performing rather comparably to the Morgan fingerprints model.

Figure 4C: There appears to still be quite some deviation in the performance of the different ensembles. Was there an ensemble stability analysis performed to identify the optimal number of models needed in the ensemble to minimize/plateau this deviation? It could be that significantly more than 10 models are needed. In other ensemble modelling techniques in the literature 50-100 models are often used.

Pg 5, lines 164-165: Although indeed convincing based on the data presented, not sure if this statement can be fully made considering polymorphism and the chance for different crystal structures with different packing arrangements.

General comment: To my experience the list of potential resolving agents that can be actually used in pharmaceutical manufacturing setting is rather limited and can be typically covered in a few HT experimental rounds. How would you use the model in this context? Would you trust the top predictions and expand the HT screen directly to the solvent space to optimize resolution? Or would you still recommend a full HT screening of resolving agents?

(Remarks on code availability)

Reviewer #3

(Remarks to the Author)

Personal disclaimer

As communicated to the Nature team before accepting to review, my background is in machine learning. These physics-based parts of the manuscript are outside of the scope of my expertise

- p. 3, ll. 82-94

- p. 5, ll. 155-163

- Supplementary Information: section 1, section 2.1, section 5

What are the noteworthy results?

The authors present an approach for the virtual screening of crystallographic enantioseparation. In my view, there can be at least three noteworthy results depending on the below conditions.

- An improvement in sensitivity of factor 4-6 over current laboratory practice, provided current laboratory practice is reflected by the benchmarks in Fig. 4A.

- New training data for the machine learning community, provided the 450 compounds selected by the authors imply a coverage of the chemical space relevant to drug discovery.

- A new combination of known approaches in chirality-aware machine learning, meaning MD-based descriptors on the one hand and a transformer encoder architecture on the other.

The new combination casts interactions between the solvent and the diastereomeric salts as cross-attentions.

Will the work be of significance to the field and related fields? How does it compare to the established literature? If the work is not original, please provide relevant references.

The authors report an improvement in sensitivity of factor 4-6 for a problem where improvements in recall (as opposed to precision) are key, described as "needle-in-the-haystack problem" by the authors.

Whether the work is of significance to the field and related fields, in my view, will depend on at least these three conditions:

- Significance for drug discovery

The authors report this improvement on a set of 450 chiral analytes.

Both this improvement and their released data set can be significant if they can show that the coverage of the chemical space is relevant to drug discovery.

Reference that illustrates chemical diversity:

Automated chiral method screening – Evaluation of generated chromatographic data sets to further optimize screening efficiency

<https://www.sciencedirect.com/science/article/pii/S0021967321002181>

- Economic significance for analytical chemistry in enantioseparation

The authors report said improvement over randomness.

Should randomness, that is an uninformed automated screening, be current practice in the screening laboratories in drug discovery, the implementation of the authors' approach can result in significant time and cost savings regarding, e.g., screening time, solvent waste, synthesis effort.

- Significance for chirality-aware machine learning

The authors report said improvement over, however, easy-to-beat benchmarks.

Their approach can be significant if they can show an improvement over the literature of chirality-aware machine learning, e.g.,

Chiral Cliffs: Investigating the Influence of Chirality on Binding Affinity

<https://chemistry-europe.onlinelibrary.wiley.com/doi/10.1002/cmdc.201700798>

Enhanced Structure-Based Prediction of Chiral Stationary Phases for Chromatographic Enantioseparation from 3D Molecular Conformations

<https://pubs.acs.org/doi/10.1021/acs.analchem.3c04028>

Retention time prediction for chromatographic enantioseparation by quantile geometry-enhanced graph neural network

<https://www.nature.com/articles/s41467-023-38853-3>

Interestingly, the proposed hybrid can imply a greater data efficiency than a transformer-only approach due to its mechanistical parts.

The significance of chirality-aware machine learning includes many fields of application, including chiral crystallization and related fields, e.g., chiral chromatography, enantioselective synthesis, structure-activity relationship modeling.

Does the work support the conclusions and claims, or is additional evidence needed?

The authors make strong claims with respect to the generalization capabilities of their approach, e.g., "Model accurately predicts chiral separation across chemical space", and in my eyes, additional evidence is needed here. This is all the more relevant given the noise and imbalance in the data as acknowledged by the authors.

I see at least two options

- weaken the claims and specify that the underlying evidence is not empirical but anecdotal on the example of chemical classes x,y,z.
- keep the claims and provide sufficient evidence on all chemical classes relevant to drug discovery.

Examples for the need of additional evidence:

- Naive and easy-to-beat benchmarks in Fig. 4A, sota benchmarks for encoding chirality are missing, see references above.

- Only 450 compounds in the data set, while there is the claim of "large-scale data".

-- How would Fig. 4C change if the authors did a scaffold split instead of a random split? A random split is the weakest pressure test for a model.

-- How well do these 450 distinct compounds cover the chemical space?

The permutations of their combinations with solvents and agents may give 2,000 distinct acid-base pairs but this does not imply a greater chemical space.

- Only 6 compounds in retrospective tests

How would Fig. 6B change if the authors picked 6 compounds from other chemical classes?

- Only 3 salts to demonstrate that attentions are meaningful

How would Fig. 5B change if the authors picked 3 salts with different scaffolds?

There is a controversy around the meaningfulness and explainability of attention maps (pro:

<https://arxiv.org/abs/1908.04626>, contra: <https://arxiv.org/abs/1902.10186>), so the

authors' claim "Attention heads in the model recognize neighboring atoms in the crystal" needs more evidence than 3 salts.

Are there any flaws in the data analysis, interpretation and conclusions? Do these prohibit publication or require revision?

I see the need for a revision. Examples that confused me:

- The authors talk about mirror images and enantiomers.

It depends on whether the mirror images are super-imposable. If they are not, it is an enantiomer.

- Statement on Morgan fingerprints (p. 4, l. 131).

It depends on the configuration of the Morgan fingerprints. If you use SMILES with stereo information, the ECFP implementation of RDKit, and set the chirality flag to true, Morgan fingerprints can capture chirality (to a limited extent, though).

MACCS cannot by design.

- Attention maps in Fig. 5B.

Typically, attentions range between 0 and 1. Here, attentions > 1. Can the authors cite the work that introduced this definition of attention?

- Column manufacturer in Supplementary Material, Table 3.

The authors list "Diacel". "Daicel" is a known manufacturer.

Is the methodology sound? Does the work meet the expected standards in your field?

I think that the reporting of data and methodology could be more detailed, not only to improve clarity but also for the reader to better understand why the authors excluded certain alternatives.

Examples:

- Transformer encoder architecture

Can the authors comment on the choice of architecture?

Transformers can outperform other architectures in high-data regimes. In low-data regimes, transformers can, however, be outperformed (<https://arxiv.org/abs/2010.11929>).

With 6000 experiments or 450 compounds, the authors are operating in a low-data regime.

- Choice of hyper parameters, Supplementary Material, Table 2

Can the authors comment on the choice of hyper parameters?

The original BERT uses up to 16 attention heads and is trained on millions of texts <https://arxiv.org/pdf/1810.04805>.

Why are 19 (cross-)attention heads suitable for 6000 experiments and how is overfitting prevented?

- Benchmarks, Fig. 4A.

-- Can the authors comment on the separation of the effect of encoding chirality from the effect of model architecture, e.g., by using the same architecture with different encodings?

-- The authors present an approach to classify enantioselective precipitation. They acknowledge the high imbalance in the data that is two-fold:

1) Most of the salts do not crystallize.

2) Most of the crystallized salts do not show an ee.

In order to demonstrate that the model can overcome both 1) and 2), can the authors comment on adding a benchmark that does only 1)?

- Designing the machine learning task (p. 4, l. 103 ff.)

Can the authors comment on the following questions?

-- Why is this procedure better than, e.g., the reverse order, meaning first classification on all data, then regression on the subset? If there is high noise, asking the model to learn bins (classification) may be more promising than asking it to learn exact values (regression).

-- How did the authors design the binning threshold? Was the threshold derived from the distribution of the data so that perturbing the data by as much as the noise level has only a minimal effect on the decision into which bin it falls?

-- Why is denoising by filtering the data better than, e.g., denoising by imposing smaller weights on the noisy data during the model training?

-- Why is this procedure better than, e.g., a multi-task model that is both a regressor and a classifier?

- Model scores

The authors prioritize recall over precision motivated by a problem that they describe as a "needle-in-the-haystack problem".

-- For a stringent argumentation that does not compare apples with pears, could streamlining the type of scores in figures, legends, text help?

sklearn https://scikit-learn.org/stable/modules/generated/sklearn.metrics.classification_report.html

is a machine learning library with great user acceptance and for binary classification, it suggests, e.g., recall, with class specifications sensitivity and specificity, and balanced accuracy.

-- The model is not balanced - precision is weak and sensitivity is stronger, s. Fig. 6B.

Can the authors comment on hyper parameter optimization with an objective function that balances recall and precision, e.g., f-score?

- Statistics, Fig. 4

Can the authors comment on the lack of uncertainty bands in Fig. 4 A, B?

I was under the impression that Fig. 4A, B shows the results from a cross-validation ensemble of 5 members, while Fig. 4C those of a bootstrap ensemble of 10 members.

- Missing explanations of N, K, L etc. in the legend of Fig. 1 in the Supplementary Material.

Is there enough detail provided in the methods for the work to be reproduced?

Parts of the approach cannot be recreated by the readers because of the lack of information.

- For recreation, the auxiliary model (auto-encoder) is needed as much as the main model (transformer encoder). Since the authors state that the training data of the auxiliary model will not be made available, the training of this model cannot be recreated. An alternative would be to publish the auto-encoder once trained, e.g., the checkpoint and an inference script to call it. Has this option been discussed with the authors?

- Details how the fingerprints were calculated, e.g., radius, size, counts or bits, are missing.

(Remarks on code availability)

It is not obvious where to find the data. Data/solvent_standartisation_dict.json seems to contain only the solvent-related data. The data could be in the repo that is given in the README file. However, when I tried to clone it, I got error 404. It would be good to know whether the other reviewers can recreate this as it could be a proxy error on my side.

Version 1:

Reviewer comments:

Reviewer #1

(Remarks to the Author)

I thank the authors for their comprehensive responses to the reviewers comments. My concerns regarding data availability and compound characterisation have been addressed and I have no further questions for the authors. I support the current version of the manuscript and SI for publication.

(Remarks on code availability)

Reviewer #2

(Remarks to the Author)

The authors have sufficiently addressed the comments made by the reviewers. In my view, the article is now suitable for publication.

(Remarks on code availability)

Reviewer #3

(Remarks to the Author)

The authors have taken considerable efforts which has improved the methodology in the machine learning part of the manuscript.

Below only the open points in ascending priority.

I suggest a second revision.

S4.1 and 4.1 claim "Model accurately predicts chiral separation across chemical space"

ChEMBL is the reference for drug-like small molecules.

It has a broader scope, e.g., it includes > 5x as many structures as GEOM-Drug

and > 120 x as many structures as the authors use from GEOM-Drug to provide evidence for their claim.

See again the reference I sent in the first review

Chiral Cliffs: Investigating the Influence of Chirality on Binding Affinity

<https://chemistry-europe.onlinelibrary.wiley.com/doi/10.1002/cmdc.201700798>

There, it takes > 800 compounds to cover the chemical space sufficiently. UMAP is used.

Can the authors comment on why, here, only 450 compounds are sufficient evidence for their claim?

I acknowledge the constraint on diastereomeric resolution but such a high discrepancy is unexpected.

Fig. 4A: The uncertainties for 3DMolCSP are so high that it happens to drop even below the most naive baseline.

What could be the reasons?

I. 177: Another difference is the model inputs.

I was under the impression that the authors input x, y, z conformer coordinates and one-hot encoding into 3DMolCSP, as opposed to MD results and COSMOtherm descriptors into the proposed model.

Coordinates are a different abstraction level than atom densities, while the models are asked to learn the same output.

Can the authors comment on how the complexity of the machine learning task depends on the abstraction level of the inputs?

Table S2, S3.4: The authors explain that the choice of 19 cross-attention heads is a result of black-box optimization. With 19 heads claimed as "optimal", it needs further explanations, in my view. 19 heads means 19 queries in the attention block, implying there are 19 distinct patterns that

a) can be identified from only 450 compounds and

b) are necessary to learn chiral separation from the data.

What are the mechanistical reasons (as opposed to treating the problem as a black box) for this high number and how much worse are the scores of models with fewer heads?

Since the authors use chiral fingerprints, there may be better options than ECFP resulting in stronger benchmarks

<https://www.x-mol.net/paper/article/1790421545364107264>

I am not sure whether "prospective" and "retrospective" borrowed from clinical studies are the best categories here as molecules are graph data, not longitudinal data.

Table S3: What does "To standard deviation" mean?

(Remarks on code availability)

I can confirm that I have access to the data

https://datadryad.org/share/SaY3Rs3klHnuKe1M47SW_vHpN9g7h4MkCXQ-anSG_wQ

Version 2:

Reviewer comments:

Reviewer #3

(Remarks to the Author)

The authors have sufficiently addressed the points made by the reviewers. In my view, the manuscript is now suitable for publication.

For the sake of completeness, here the correct reference

<https://www.sciencedirect.com/science/article/abs/pii/S0021967321002181>, Fig. 6.

Apologies for sending an earlier paper of the same authors, <https://chemistry-europe.onlinelibrary.wiley.com/doi/10.1002/cmdc.201700798>.

(Remarks on code availability)

Response to the reviewers

We appreciate the reviewers’ time and feedback, which has significantly strengthened our manuscript. We are encouraged by the reviewers’ recognition of the practical importance of our work in modelling diastereomeric resolutions.

Reviewers identified two main areas for improvement: experimental data transparency and further clarification of our machine learning methodology and validation. In response to these constructive comments, we have implemented the following key revisions:

- **Enhanced Data Accessibility and Experimental Detail:** Requested experimental data, including CIFs, chiral HPLC traces, and detailed experimental protocols are now included in the Supplementary Information. We have clarified yield calculation procedures, experimental scale, and enantiomer assignment methods. A data availability statement with a direct link to the dataset has been added to the main text.
- **Expanded Model Benchmarking:** To provide a more rigorous evaluation, we have benchmarked our model against a state-of-the-art, chirality-aware method for chromatographic enantioseparation. This comparison is detailed in the revised manuscript and Supplementary Information.
- **Refined Attention Analysis Interpretation:** Section 4.2 is now clarified to emphasize the limitations of directly interpreting attention head weights as having deep physical meaning, presenting this analysis as a sanity check for chemically meaningful spatial relationships.
- **Detailed Computational Methods in SI:** Expanded Supplementary Information sections now provide comprehensive details on computational methods, including hyperparameter tuning and an analysis of the dataset’s chemical diversity.
- **General Clarity and Nuance Improvements:** Multiple smaller revisions have been made throughout the manuscript to improve clarity, and several sections have been expanded to include more nuanced discussions of model limitations, assumptions, and future directions.
- **Further exploration of model architecture:** We acknowledge the potential for improvement but believe such investigations are beyond the scope of the current proof-of-concept study, particularly given the positive prospective validation results. However, we now mention some of these future research directions at the conclusion of the manuscript.

Finally, we note that we have significantly increased the speed of the MD simulations needed to generate the representations by switching to a classical force field. With this change we can more thoroughly sample the conformational landscape of the acid-base pair. The models retrained on these representations have further improved the results on retrospective testing, all while having a lower computational cost.

We believe these revisions comprehensively address the reviewers’ core concerns, resulting in a more robust and clearly communicated manuscript.

Reviewer 1

This paper by Lee and coworkers focuses on the use of molecular dynamics and training a neural network to aid in the prediction of diastereomeric salt formation for chiral resolutions. A previously unreleased historical dataset of 6,000 resolution experiments forms the training set and demonstrates the difficulty in current screening for diastereomeric salt resolutions. Even with setting a very generous bar for “success” (>20% solid mass fraction and >25% ee), only 3% of the historical dataset falls in this category; highlighting the difficulty in the classical screening approach. Once built, Lee and coworkers use their model to resolve 3 racemates with higher success rate than the original dataset without the use of the model. This predictive approach will be of interest to those seeking to access enantioenriched materials. Particularly since diastereomeric resolutions are still commonplace within the pharmaceutical industry. However, key points need to be addressed before publication of this work.

Reviewer Point P 1.1 — I was very interested in the 6,000 resolution experiment dataset that was reportedly used in this work, however, I cannot find this dataset within the SI or the “additional data” zip file. I believe it would be beneficial to make this dataset visible in the SI, perhaps as a table, so that chemists can access it

Reply: We apologize that the link to the raw dataset was not more prominent in the initial submission. We have now updated the main text to include a data availability statement with a clear link to the dataset.

Reviewer Point P 1.2 — Did the neural network use any crystallographic data during training?

Reply: No, the neural network did not use any crystallographic data during training. This is stated in the conclusion of Section 4.2:

“ However, we find it encouraging that our model - **without explicit crystallographic training** - qualitatively captures chemically meaningful spatial relationships. ”

Reviewer Point P 1.3 — Figure 5C shows an X-Ray crystal structure. Has this data been collected by the authors? If it is an original crystal structure reported as part of the paper, the CIF needs to be deposited on the CCDC and the deposition number should be supplied for reviewing. If it is from a previously published structure, it requires a reference.

Reply:

Thank you for raising this concern. We have now deposited these crystal structures and include the relevant deposition numbers in the SI Section S7.2.

Reviewer Point P 1.4 — Pg 5, Line 151-152 mentions 3 more crystal structures. These need to be described within the SI and deposited into the CCDC. The deposition numbers are required for reviewing the obtained data.

Reply:

The three crystal structures referred to in that section are the same ones as in Figure 5, which we have deposited as indicated in our response to Point P 1.3.

In addition, we have deposited the crystallised salts from the prospective experiment. All deposition numbers are now included in the SI.

Reviewer Point P 1.5 — Figure 6C – I’m not sure what new information this chart is conveying. The “needle-in-the-haystack” problem was already shown in Figure 2. The focus should be on demonstrating the resolutions predicted by the model instead of the historical dataset.

Reply:

Thank you for this suggestion. We have moved the panel in question to the SI. We believe the figure is still useful as we use it to justify the exact threshold of considering a resolution positive during model training. This is detailed in Section S4.3.

Reviewer Point P 1.6 — Pg 7, Line 171 mentions 6 racemates tested – 3 acids and 3 bases. How did the other acid and 2 bases perform that are not shown in Figure 6?

Reply: The other three racemates did not reach the success criteria with any of the conditions prospectively tested. To make this explicit, we have added these structures to Figure 6.

Reviewer Point P 1.7 — How was the configurational assignment of the resolved materials performed? Did the model predict the correct enantiomer?

Reply:

In the training set we did not include enantiomeric configuration as this was rarely known at the time of screening and hence not captured in our dataset. During prospective testing, however, the configurations of the top hit for each successful racemate were determined by X-ray crystallography which allowed accompanying hits for those racemates to be determined also.

With regards to the model predicting the correct enantiomer – the model, by design, does not predict which diastereomeric salt precipitates. We constructed the architecture to be agnostic to the exact enantiomer of the resolving agent employed, which is what determines which enantiomer of the original racemate precipitates and which remains in solution.

Reviewer Point P 1.8 — Upon checking the SI, there are no obtained characterization data for the materials obtained after the resolution so there is no verification of the identity of any of the claimed compounds - No NMR spectra, chiral HPLC chromatograms, MS, IR, melting points, optical activity (optical rotation or specific rotation) or crystal structures are reported to prove the identity and purity of the resolution experiments. At a minimum the paper will require chiral HPLC (to prove ee) and ¹H NMR spectra (to prove identity and purity) and HRMS or crystal structures (to prove elemental composition). Currently, there is no experimental data in the manuscript or SI to show that the resolutions were successful.

Reply:

For the prospective tests we have included in the SI chiral SFC-MS spectra of a representative sample using selected ion monitoring (SIM) mass spectroscopy on the most relevant mass ion to assess % e.e. and % mass fraction in solution, as well as the full M/S spectra for the major enantiomer peak which match expected fragmentation patterns. The mass ion used is given in the peak label of the chromatogram.

X-ray crystallographic data are also provided for the three successfully resolved racemates, showing the diastereomeric salts with their best resolving agent and confirming the starting material structures themselves as well as the resolved enantiomer configuration. We hope that this satisfies the characterization requirements for prospective testing.

Sadly, our historical training dataset (pulled from an archival database) does not include characterization spectra, hence we are unable to provide such.

Reviewer Point P 1.9 — The resolution experiments are conducted on a very small scale (0.04 mmol scale). For example, the 2-phenylmorpholine resolution with acetyl leucine, with a 45% yield, it is expected to yield 6 mg of material. On this small scale (and without ¹H NMR spectra to verify) it is possible to get unrepresentative yields due to the presence of trace solvents. It would be preferable to perform these resolutions on a 0.5-1.0 mmol scale, to get more representative yields and supply characterization data to verify the solids are dry when calculating yields from their obtained masses (or use an appropriate assay method, such as NMR assay with an internal standard).

Reply:

We completely agree that these concerns would be valid for **gravimetrically** measured yields on such small scales.

In our case, we believe that our procedure avoids this pitfall as rather than gravimetric quantification of the solids for our hits, we rinse and then dissolve the solids to a known volume for analysis by calibrated SFC-MS.

Our process is as follows: at the end of the reaction we centrifuge the reaction mixture and remove the crystallization liquors by pipette. We then rinse the solids in fresh solvent (the same solvent as the crystallization was run in), centrifuge and remove the rinse liquors (combining with reaction liquors), and finally dissolve the rinsed solids to a known volume with methanol for analysis. We then use SFC-MS area versus single-point calibration for quantification, running all samples in triplicate to reduce error. We also dilute the combined liquors from the reaction and solids rinse with methanol to the same known volume and analyse those by SFC-MS also.

We believe that this provides accurate quantification for the hits. We have edit the experimental procedure in SI Section S7.1 to make this point more explicit.

Reviewer Point P 1.10 — As such, these points require addressing before publication. Most importantly, key experimental data has not been collected (characterization of compounds) or made available for the review process (no chiral HPLC traces of resolved materials, X-ray structure CIFs/CCDC deposition numbers, masses of obtained materials).

Reply:

We appreciate the honest feedback. The SI has been updated with the missing experimental data, including crystal deposition numbers and chiral SFC-MS traces. With these changes, we believe our results now better supported by the data.

Minor

Reviewer Point P 1.11 — Pg1, Line 37 – Replace “acidic/basic centre” with “acidic/basic functional group”

Reply: Fixed.

Reviewer Point P 1.12 — Pg1, Line 38 – Replace “insoluble salt” with “less soluble diastereomeric salt”

Reply: Fixed.

Reviewer Point P 1.13 — Figure 2 – the 5% error line for the “maximum possible EE” is only visible when figure is magnified – perhaps the use of higher contrast colour or a black dotted line would make it more visible to the reader.

Reply: Fixed.

Reviewer Point P 1.14 — Figure 6 switches to using %ee for reporting the enantioenrichment for the resolutions – perhaps it would be worth using % enantiomeric excess label in the figure.

Reply: Fixed, and the figure in question has been moved to SI as Figure S5

Reviewer 2

The authors have developed a physics-informed machine learning model to predict resolving agents for the diastereomeric classical resolution of chiral molecules, utilizing a proprietary dataset of 6,000 prior experimental resolution attempts. Although, as acknowledged by the authors, the dataset is biased, with significantly more 'failed' than 'successful' attempts, the modelling methodology is rigorous and carefully designed, and the paper is exceptionally well-written. I only have a few minor comments for consideration (not in order of importance):

Reviewer Point P 2.1 — Abstract line 15 and pg. 2 line 46: 4-6 times more accurate than what? The benchmark only becomes clear much later in the text, please consider adding something here for the abstract readers.

Reply: We have clarified the sentence to read “ In retrospective tests, our approach reaches a four to six-fold improvement over the historical - trial and error based - hit rate. ”

Reviewer Point P 2.2 — Figure 2: The graph shows that 97% of the test cases ended up being unsuccessful. In my view, a key question is also why is this the case? While indeed the solubility differences of the diastereomeric salt pairs drive effective separation (main scope of this work), there are other reasons why a resolution could fail. For example, low ee could mean that the diastereomeric salts never formed. Other possibilities are diastereomeric double salts, solid solutions, or stability issues. More details on the training dataset could help here.

Reply: This is exactly right – we are assuming that all experiments have reached thermodynamic equilibrium, when in fact for some acid/base pairs might have very slow crystallisation kinetics, among other factors you have mentioned. Unfortunately, the high throughput nature of our data do not provide such fine grained details about the experiments.

We have expanded the discussion in Section 2 to include a more nuanced discussion of these issues:

However, in practice, the ideal resolution is seldom attained: Figure 2 shows that most of the data lie in the low mass fraction, low enantiomeric excess region with 97% of the data having either m.frac. lower than 20% or e.e. lower than 25%, or both. **A variety of factors can contribute to these outcomes. For instance, a low mass fraction could arise from slow crystallisation kinetics, where experiments are terminated before complete crystallisation, or from an inappropriate solvent choice where both diastereomeric salts remain dissolved. Similarly, a low enantiomeric excess, even with sufficient solid material, can be attributed to phenomena like the formation of solid solutions or mixed salts. Given the high-throughput nature of our data collection, detailed investigation of each experiment is not feasible. Therefore, for further modelling purposes, we operate under the assumption that all crystallisations have reached thermodynamic equilibrium and that the relative stability of the two diastereomeric salts is the primary drivers of resolution success.**

Reviewer Point P 2.3 — Pg. 7, line 175 and data in Figure 2: Linked to the previous comment. Could you please add (probably in the SI) the list of chiral resolving agents, substrates (if possible) and conditions used in the virtual and experimental screenings? From the experimental screenings, do you see any obvious patterns in the use of certain resolving agents vs success of resolution?

Reply: Yes, we provide the full list of chiral resolving agents, substrates, and the conditions for the historic data, and both retrospective and prospective resolutions. We apologize for not making these data more prominent during review; we have now added a Data Availability statement in the main text with a link to all of the data.

Reviewer Point P 2.4 — Figure 2: In classical resolution, it is often found that often < 1 equiv of resolving agent yields the most effective resolution conditions. What conditions did the resolution experiments in fig 2 use in terms of resolving agent equiv? Also, is this effect something that the model could in theory capture?

Reply:

We agree that using less than 1 equivalent of resolving agent can often lead to more effective resolution, as seen in methods like "half equivalent" resolutions. However, to maintain consistency across the high-throughput screening experiments, which are depicted in Figure 2, 1 equivalent of resolving agent was used.

It's standard practice that when a promising hit is identified from such a screen, further optimization is performed. This optimization often includes varying the ratio of racemate to resolving agent. These targeted optimisation data are not included in the current data release and were not used for model training.

In principle, our model could learn to capture the effects of different molar ratios if trained on a dataset that includes resolution experiments performed across a range of ratios. Exploring this aspect would be an interesting direction for future model development and data collection.

Reviewer Point P 2.5 — Pg. 3 line 80: Relative solubility of the diastereomeric salt pairs will also depend on polymorphism and solvates. While looking at the different resolving agents is indeed the first step, solvent effects should not be neglected. The presented model cannot capture such effects, but it is worth mentioning.

Reply:

Thank you for highlighting the importance of polymorphism and solvate formation. We agree that these factors can significantly influence diastereomeric salt solubility and resolution outcomes. We have incorporated this crucial context into the manuscript, as you suggested.

Resolutions fundamentally depend on the energy differences between these diastereomeric pairs, which influence their relative lattice and solvation energies, and consequently, their relative solubility. **Of course, the relationship between diastereomeric salt pairs and their solubility is not always straightforward. Phenomena such as polymorphism and solvate formation mean that each diastereomeric salt pair can exist in multiple solid forms, each with potentially different energies. Despite this added complexity, we hypothesise that by directly representing the acid-base pairs, our machine learning model can still effectively learn the key physical differences driving successful resolution, even without explicitly accounting for polymorphism and solvate effects.**

Reviewer Point P 2.6 — Figure 4A: It seems a bit surprising that the “zero knowledge” model is performing rather comparably to the Morgan fingerprints model.

Reply:

“Zero knowledge” models can be surprisingly effective when the training data has some underlying bias. If certain resolving agents are consistently effective across multiple racemates within the training set, the “zero knowledge” model, despite its simplicity, can quickly exploit these patterns.

This observation also directly addresses the question you raised in Point P 2.3. The fact that the “zero knowledge” model performs poorly in that specific context indicates that there isn’t a single resolving agent that is universally effective across all racemates in our dataset.

Regarding the Morgan fingerprint model, while it does encode chemical structure information and some aspects of chirality, it is inherently limited in capturing the more complex chiral interactions relevant to resolution. The comparable performance of the “zero knowledge” model and the Morgan fingerprint model underscores that a more sophisticated approach, capable of capturing these nuanced chiral effects, is necessary for this task.

Furthermore, in answering Reviewer 3’s comments, we now include a different neural network in our comparison. As shown in the updated figure, this more complex model outperforms both the “zero knowledge” and Morgan fingerprint baselines but still falls short of our approach. We believe this difference in performance further highlights the importance of our pair-representation in capturing the effects of chirality.

Reviewer Point P 2.7 — Figure 4C: There appears to still be quite some deviation in the performance of the different ensembles. Was there an ensemble stability analysis performed to identify the optimal number of models needed in the ensemble to minimize this deviation?

Reply:

We did not perform a formal ensemble stability analysis. However, we believe the relatively large variance is primarily due to data variability, not model instability.

We have added Section S4.2 in the SI, showcasing how some racemates have over a thousand experiments associated with them, while many others have less than a hundred. When generating Figure 4C, the training data for each model in the ensemble was resampled, leading to high variability.

Reviewer Point P 2.8 — Pg 5, lines 164-165: Although indeed convincing based on the data presented, not sure if this statement can be fully made considering polymorphism and different crystal structures with different packing arrangements.

Reply:

Thank you for raising this issue. Your comments and similar concerns raised by Reviewer 3 have lead us to temper the claims we have made in Section 4.2.

We now conclude the Section as such:

While the interpretability of attention weights remains a subject of ongoing debate [1, 2], we find it encouraging that our model, without explicit crystallographic training, qualitatively captures chemically meaningful spatial relationships. The observed tendency to assign higher attention to atom pairs that are close in the crystal structure provides a valuable sanity check, suggesting the model is sensitive to interactions relevant to molecular packing.

We believe the claim that the model is **sensitive** to interaction relevant to molecular packing is well supported given both the analysis we make in Section 4.2 and the results of the prospective experiment.

Reviewer Point P 2.9 — General comment: The list of potential resolving agents is often rather limited in pharmaceutical manufacturing. How would you use the model in this context? Would you trust the top predictions and expand the screen, or would you still recommend a full HT screening?

Reply:

Thank you for your comment regarding the practical application of our model in pharmaceutical manufacturing. This is an important point.

We see two main practical uses for the model in this context.

First, the model can improve the efficiency of HT screening by filtering out resolving agents that are likely to fail. Screening is usually done on a plate-by-plate basis. Ideally, with the help of our model the number of plates needed to find a hit would be one for all compounds. Of course, this is likely to optimistic, in which case, with sufficient resources, the model could be updated with each batch of new data yielding an iterative experimental design.

Second, the model can guide the expansion of routinely used resolving agents. As you noted, resolving agent lists can be limited. We believe there is an untapped resource of single enantiomer chiral acids and bases. Our model can identify promising candidates from this broader chemical space, suggesting novel resolving agents for synthesis and testing, thus broadening the available tools for pharmaceutical manufacturing.

In short, we hope the model can both increase efficiency of current HT workflows, and guide the search for yet untapped resolving agents.

Reviewer 3

Personal disclaimer

As communicated to the Nature team before accepting to review, my background is in machine learning. These physics-based parts of the manuscript are outside of the scope of my expertise:

- p. 3, ll. 82-94
- p. 5, ll. 155-163
- Supplementary Information: section 1, section 2.1, section 5

What are the noteworthy results?

The authors present an approach for the virtual screening of crystallographic enantioseparation. In my view, there can be at least three noteworthy results depending on the below conditions.

- An improvement in sensitivity of factor 4-6 over current laboratory practice, provided current laboratory practice is reflected by the benchmarks in Fig. 4A.
- New training data for the machine learning community, provided the 450 compounds selected by the authors imply a coverage of the chemical space relevant to drug discovery.
- A new combination of known approaches in chirality-aware machine learning, meaning MD-based descriptors on the one hand and a transformer encoder architecture on the other. The new combination casts interactions between the solvent and the diastereomeric salts as cross-attentions.

Will the work be of significance to the field and related fields? How does it compare to the established literature? If the work is not original, please provide relevant references.

The authors report an improvement in sensitivity of factor 4-6 for a problem where improvements in recall (as opposed to precision) are key, described as "needle-in-the-haystack problem" by the authors. Whether the work is of significance to the field and related fields, in my view, will depend on at least these three conditions:

- **Significance for drug discovery**

The authors report this improvement on a set of 450 chiral analytes. Both this improvement and their released data set can be significant if they can show that the coverage of the chemical space is relevant to drug discovery. Reference that illustrates chemical diversity: Automated chiral method screening – Evaluation of generated chromatographic data sets to further optimize screening efficiency <https://www.sciencedirect.com/science/article/pii/S0021967321002181>

- **Economic significance for analytical chemistry in enantioseparation.**

The authors report said improvement over randomness. Should randomness, that is an un-informed automated screening, be current practice in the screening laboratories in drug discovery, the implementation of the authors' approach can result in significant time and cost savings regarding, e.g., screening time, solvent waste, synthesis effort.

- **Significance for chirality-aware machine learning**

The authors report said improvement over, however, easy-to-beat benchmarks. Their approach can be significant if they can show an improvement over the literature of chirality-aware machine learning, e.g.,:

- Chiral Cliffs: Investigating the Influence of Chirality on Binding Affinity <https://chemistry-europe.onlinelibrary.wiley.com/doi/10.1002/cmdc.201700798>
- Enhanced Structure-Based Prediction of Chiral Stationary Phases for Chromatographic Enantioseparation from 3D Molecular Conformations <https://pubs.acs.org/doi/10.1021/acs.analchem.3c04028>
- Retention time prediction for chromatographic enantioseparation by quantile geometry-enhanced graph neural network <https://www.nature.com/articles/s41467-023-38853-3>

Interestingly, the proposed hybrid can imply a greater data efficiency than a transformer-only approach due to its mechanistical parts. The significance of chirality-aware machine learning includes many fields of application, including chiral crystallization and related fields, e.g., chiral chromatography, enantioselective synthesis, structure-activity relationship modeling.

Reply: Thank you for highlighting these three aspects of our work; we agree that they are the most significant contributions.

Does the work support the conclusions and claims, or is additional evidence needed?

The authors make strong claims with respect to the generalization capabilities of their approach, e.g., "Model accurately predicts chiral separation across chemical space", and in my eyes, additional evidence is needed here. This is all the more relevant given the noise and imbalance in the data as acknowledged by the authors.

I see at least two options a) weaken the claims and specify that the underlying evidence is not empirical but anecdotal on the example of chemical classes x,y,z. b) keep the claims and provide sufficient evidence on all chemical classes relevant to drug discovery.

Examples for the need of additional evidence:

Reviewer Point P 3.1 — Naive and easy-to-beat benchmarks in Fig. 4A, sota benchmarks for encoding chirality are missing, see references above.

Reply: We acknowledge the point about the need for a more challenging benchmark.

We note that none of the provided references are directly applicable to diastereomeric salt resolution; therefore, we have modified one of the models (3DMolCSP) and trained it ourselves to enable a fair, apples-to-apples comparison.

We have added Section S5 where we show that our approach beats 3DMolCSP across all the metrics mentioned in Point P 3.13 bar one. We include the performance of 3DMolCSP in the updated Figure 4A. Section S5 also explains the modifications made to the original model.

Reviewer Point P 3.2 — Only 450 compounds in the data set, while there is the claim of "large-scale data".

Reply: We understand the comment regarding the number of unique compounds (450) and appreciate the opportunity to clarify our use of "large-scale." The value of our data lie not only in the number of unique compounds but also in the physical experiments that were ran with them.

Each diastereomeric resolution experiment, even with the same acid-base pair, is a distinct data point that carries new information. Similar to natural language datasets where size is measured by

text instances, not just unique words, our dataset's richness comes from these diverse experimental conditions.

We acknowledge that within the broader machine learning community, 6,000 reactions may not be "large-scale" in absolute terms. Therefore, we have revised the manuscript to describe our dataset as "the largest diastereomeric salt crystallization dataset released to date," more accurately reflecting its scope within this specific domain.

Reviewer Point P 3.3 — How would Fig. 4C change if the authors did a scaffold split instead of a random split? A random split is the weakest pressure test for a model.

Reply: We agree that a random split is a relatively weak test of a model's generalization capabilities. That is why we used a scaffold-split test design for the main retrospective experiment. The results from these scaffold-split experiments are shown in Figure 4A and B, and in the new Section S5 in the SI. Moreover, the final validation of our model is done in a prospective experiment using racemates not seen during training as described in Section 4.3.

The purpose of Figure 4C is specifically to demonstrate the scaling behaviour of our approach with increasing amounts of training data, rather than to assess its predictive performance for specific scaffolds. A scaffold split would likely introduce significant noise into the learning curve due to the unbalanced distribution of scaffolds within the data, which we show in the new Section S4.2 in the SI.

Reviewer Point P 3.4 — How well do these 450 distinct compounds cover the chemical space? The permutations of their combinations with solvents and agents may give 2,000 distinct acid-base pairs but this does not imply a greater chemical space.

Reply: We appreciate the question about our compounds' chemical space coverage. As detailed in the new Section S4, these 450 compounds originate from pharmaceutical projects and are intentionally drug-like, representing chemical space relevant to drug discovery.

To quantify diversity, we compared our dataset against the GEOM-Drug collection. Figures S3A and B show our dataset has comparable distributions of mean similarity and QED scores to GEOM-Drug. PCA of Morgan fingerprints (Figure S3C) demonstrates our compounds sample high-density regions of drug-like chemical space, with the convex hull of our dataset encompassing approximately 70% of GEOM-Drug samples.

While the nature of diastereomeric resolutions and the dataset size limit the extent of explored chemical space, these analyses show our dataset is diverse and representative within pharmaceutical chemical space.

Reviewer Point P 3.5 — Only 6 compounds in retrospective tests How would Fig. 6B change if the authors picked 6 compounds from other chemical classes?

Reply: We believe there may be a misunderstanding regarding the nature of the experiment depicted in Figure 6. This figure presents the results of a **prospective** validation, not a retrospective analysis. As explicitly stated in the main text:

To test the model in a production-like scenario, we selected **new unresolved chiral acids and bases** and used the model to identify the resolving agents most likely to yield successful resolutions.

The final experimental design involved six diverse racemates, three acids and three bases, **which were not present in the original data.**

Therefore, the six compounds in Figure 6 were intentionally chosen from outside our training dataset, both as racemates and as resolving agents. This prospective approach was specifically designed to evaluate the model's ability to generalize to novel compounds, which is a more rigorous test than a retrospective analysis.

Reviewer Point P 3.6 — Only 3 salts to demonstrate that attentions are meaningful.

How would Fig. 5B change if the authors picked 3 salts with different scaffolds? There is a controversy around the meaningfulness and explainability of attention maps (pro: <https://arxiv.org/abs/1908.04626>, contra: <https://arxiv.org/abs/1902.10186>), so the authors' claim "Attention heads in the model recognize neighboring atoms in the crystal" needs more evidence than 3 salts.

Reply: Thank you for bringing these helpful references to our attention. We acknowledge that our attention analysis is based on a limited number of crystal structures. While we have now included an additional crystal in the analysis, bringing the total to four, we understand that this still does not constitute a statistically meaningful sample size.

Expanding this analysis to include more crystal pairs is unfortunately unfeasible. Collecting X-ray diffraction data is not routinely performed for all racemates in our dataset, and even when available, it is typically only obtained for the single best crystallization conditions.

To address your concerns regarding interpretability and sample size, we have reworded Section 4.2 to be more nuanced and to clearly highlight the inherent limitations of such analyses. Specifically, we now conclude the section with the following statement:

While the interpretability of attention weights remains a subject of ongoing debate [1, 2], we find it encouraging that our model, without explicit crystallographic training, qualitatively captures chemically meaningful spatial relationships. The observed tendency to assign higher attention to atom pairs that are close in the crystal structure provides a valuable sanity check, suggesting the model is sensitive to interactions relevant to molecular packing.

Reviewer Point P 3.7 — Statement on Morgan fingerprints (p. 4, l. 131).

It depends on the configuration of the Morgan fingerprints. If you use SMILES with stereo information, the ECFP implementation of RDKit, and set the chirality flag to true, Morgan fingerprints can capture chirality (to a limited extent, though). MACCS cannot by design.

Reply: We confirm that we used the ECFP fingerprints as implemented in RDKit, and that we used SMILES strings with stereo information and the chirality flag set to true.

Section S5.2 now reads: "Morgan fingerprints were generated with RDKit using a chirality flag set to True, a radius of 2 (equivalent to ECFP4), and 1024 bits."

Reviewer Point P 3.8 — Attention maps in Fig. 5B.

Typically, attentions range between 0 and 1. Here, attentions > 1. Can the authors cite the work that introduced this definition of attention?

Reply: The attention maps indicate not the weights themselves, but their ratio to the uniform attention baseline. The attention weights themselves were always within the range of 0 and 1.

Attentions weights themselves are not a good choice for a clear figure, as they are not an intensive property. Larger molecules will have lower attentions simply because they are spread over more atoms. On the other hand, the ratio does not depend on the total number of atoms and effectively highlights pairs that are prioritised by the attention mechanism.

To avoid any confusion, we have added an additional label to Figure 5 stating the the heatmap shows attention ratios.

Reviewer Point P 3.9 — Transformer encoder architecture

Can the authors comment on the choice of architecture? Transformers can outperform other architectures in high-data regimes. In low-data regimes, transformers can, however, be outperformed (<https://arxiv.org/abs/2010.11929>). With 6000 experiments or 450 compounds, the authors are operating in a low-data regime.

Reply: Our selection of the Transformer architecture was driven by its suitability for the specific challenges of our task and dataset:

1. **Atom-Centred Representation Compatibility:** A core element of our work is the atom-centred representation of acid-base pairs. Transformers are inherently well-suited to process such set-like inputs, allowing us to fully leverage this representation.
2. **Long-Range Interaction Capture:** Predicting diastereomeric crystal energy necessitates capturing intricate non-bonded interactions. The Transformer architecture’s attention mechanism excels at modeling these long-range dependencies.
3. **Established Cheminformatics Success:** We prioritized an architecture with a demonstrated track record in cheminformatics. Transformers have shown promise in various related applications, lending confidence to our choice.

We have also expanded the main text to mention these factors directly.

While we recognize your point about potential limitations of Transformers in low-data regimes, we believe the architecture’s strengths align well with the specific demands of our problem. Furthermore, we hypothesize that for this dataset, exploring transfer learning from larger chemical datasets would likely yield more substantial improvements than further architectural refinements alone.

Reviewer Point P 3.10 — Choice of hyper parameters, Supplementary Material, Table 2

Can the authors comment on the choice of hyper parameters?

The original BERT uses up to 16 attention heads and is trained on millions of texts <https://arxiv.org/pdf/1810.04805>. Why are 19 (cross-)attention heads suitable for 6000 experiments and how is overfitting prevented?

Reply: Hyper parameter selection, including the number of attention heads, was optimised using Optuna. We systematically explored different hyper parameter combinations, evaluating performance using binary cross-entropy loss on a dedicated validation set during retrospective testing. We added a description of the hyper parameter tuning process in Section S3.4.

Regarding the number of attention heads: while it is true that BERT has less attention heads than our model, the risk of overfitting is primarily governed by the overall model parameter count, not solely by the number of attention heads in isolation. Our model, with 8.5 million parameters, is substantially

smaller than BERT (340 million parameters), resulting in a significantly lower capacity to overfit the training data, even with 19 attention heads.

Furthermore, we used dropout during training, which provides an additional mechanism to prevent overfitting. The combination of a relatively small parameter count, dropout regularization, and hyperparameter tuning via validation set performance allowed us to train the model without signs of overfitting, despite the seemingly large number of attention heads compared to the dataset size.

Reviewer Point P 3.11 — Benchmarks, Fig. 4A.

- Can the authors comment on the separation of the effect of encoding chirality from the effect of model architecture, e.g., by using the same architecture with different encodings?
- The authors present an approach to classify enantioselective precipitation. They acknowledge the high imbalance in the data that is two-fold: 1) Most of the salts do not crystallize. 2) Most of the crystallized salts do not show an ee. In order to demonstrate that the model can overcome both 1) and 2), can the authors comment on adding a benchmark that does only 1)?

Reply: We thank the reviewer for raising these important questions about our benchmark experiments.

Regarding the separation of encoding and model architecture, we agree that further dissecting the contribution of each component could be a valuable research direction. However, we believe that the existing benchmarks in Figure 4A already provide strong evidence for the effectiveness of our **combined** approach.

As commented earlier, we have included a state-of-the-art deep learning method, 3DMolCSP, in our benchmark. The expanded benchmark now covers a range of approaches: from simple models with simple representations (Random Forest with "zero knowledge" and Morgan fingerprints), to a complex model with a learned representation (3DMolCSP), and finally our approach using a complex model with a physics-based representation.

It's important to emphasize that our innovation lies not in the atom-density encoding itself – which is well-established in the field of machine learning force fields – but in its application to molecule **pairs** and its use for the specific problem of diastereomeric resolutions.

Regarding the suggestion to benchmark crystallization yield prediction in isolation, we acknowledge that this is an interesting area for investigation. Indeed, our regression experiments suggest that enantiomeric enrichment prediction is often more accurate than solid yield prediction. However, the core objective of this work is to address the challenge of **enantioselective** crystallization. The existing benchmarks aim to directly demonstrate the value of our approach for its intended application. Pursuing a separate crystallization-only benchmark, while potentially interesting, is outside the primary scope of this paper and would not fundamentally alter our main conclusions about predicting chiral resolution success.

Reviewer Point P 3.12 — Designing the machine learning task (p. 4, l. 103 ff.)

- Why is this procedure better than, e.g., the reverse order, meaning first classification on all data, then regression on the subset? If there is high noise, asking the model to learn bins (classification) may be more promising than asking it to learn exact values (regression).

- How did the authors design the binning threshold? Was the threshold derived from the distribution of the data so that perturbing the data by as much as the noise level has only a minimal effect on the decision into which bin it falls?
- Why is denoising by filtering the data better than, e.g., denoising by imposing smaller weights on the noisy data during the model training?
- Why is this procedure better than, e.g., a multi-task model that is both a regressor and a classifier?

Reply: Thank you for your insightful questions regarding the design of our machine learning task. We will address each of your points directly below to clarify our choices.

- **Regression followed by classification vs. classification followed by regression:** We chose regression first, then classification, because empirically this order worked better for our task. As you suggested, we explored classification followed by regression in reverse order, and we've added a new Section S5 in the supplementary information to show this comparison. For convenience, we include the table showing these comparisons below.

Our intuition is that performing regression initially allows the network to extract richer information from the data. Subsequently, framing the second step as classification helps the network learn a scoring function specifically designed to prioritize resolution conditions, which is our primary objective. In contrast, if regression is the second task we need to use a hand-crafted figure of merit to rank the conditions.

- **Binning threshold design:** The mass fraction threshold was selected based on the advice of experimental colleagues. For the enantiomeric enrichment threshold, we aimed for the highest value that still provided a sufficient number of positive examples for a balanced classification task. See Figure S5 in the SI, which shows that this choice also coincides with a large cliff in the figure of merit distribution.
- **Denoising by filtering vs. weighted training:** Regarding denoising, we actually use weighted sampling during training, as detailed in Section S3.3, which effectively down-weights noisier samples.

To facilitate training, we used weighted sampling to construct batches during training. For classification, each batch of data was constructed to contain an equal number of successful and unsuccessful resolutions. For regression, each batch of data contains an equal number of samples from three partitions: $X < 33\%$, $33\% < X < 66\%$, and $X > 66\%$, where X is either mass fraction of the solid or the enantiomeric excess. In our experiments, weighted sampling significantly improved the rate of convergence during training and the overall performance of the model.

- **Comparison to multi-task models:** We acknowledge that exploring alternative training approaches, such as a multi-task or Siamese/Contrastive learning, could lead to further performance improvements. Even larger gains might be made if a suitable data/task combination for transfer learning could be found. However, a complete exploration of model architecture and learning task is outside the scope of the current work but we have highlighted some of these exciting directions in the conclusion.

Method	Positive Class Metrics			Negative Class Metrics			Overall Metrics		
	F1	Precision	Recall	F1	Precision	Recall	Bal. Acc.	AUC ROC	p -thresh.
Baselines									
Random Encoding	.03± .01	.04± .02	.02± .01	.938± .001	.923± .001	.953± .003	.488± .004	.45± .01	.047± .007
Morgan Encoding	.08± .02	.10± .02	.06± .01	.941± .001	.926± .001	.956± .002	.508± .006	.45± .01	.046± .004
Our Approach									
Direct Classification	.07± .02	.09± .02	.05± .01	.940± .001	.925± .001	.955± .001	.505± .007	.47± .09	.22± .05
Direct Regression	.17± .03	.23± .04	.14± .02	.947± .002	.932± .002	.962± .002	.55± .01	.51± .01	.016± .005
Finetune Classification from Classification	.07± .01	.09± .01	.054± .008	.940± .001	.925± .001	.955± .001	.505± .004	.45± .05	.35± .09
Finetune Classification from Regression	.23± .05	.31± .06	.19± .04	.951± .003	.936± .003	.966± .003	.58± .02	.61± .06	.4± .1
Finetune Regression from Classification	.05± .01	.07± .02	.04± .01	.939± .001	.924± .001	.955± .001	.498± .006	.45± .04	.09± .02
Finetune Regression from Regression	.20± .03	.26± .04	.16± .02	.948± .002	.933± .002	.964± .002	.56± .01	.52± .01	.043± .006
3DMolCSP									
Direct Regression	.14± .06	.19± .08	.11± .05	.945± .004	.930± .004	.960± .004	.54± .03	.58± .06	.06± .01
Direct Classification	.05± .01	.06± .01	.036± .008	.939± .001	.924± .001	.954± .001	.495± .004	.49± .04	.3± .2
Finetune Regression from Regression	.08± .07	.10± .09	.06± .05	.941± .004	.926± .004	.956± .004	.51± .03	.67± .05	.14± .04
Finetune Classification from Classification	.08± .01	.10± .02	.06± .01	.941± .001	.926± .001	.956± .001	.508± .006	.57± .04	.74± .06
Finetune Regression from Classification	.07± .04	.09± .05	.05± .03	.940± .003	.925± .003	.955± .003	.51± .02	.59± .05	.24± .05
Finetune Classification from Regression	.10± .01	.13± .02	.08± .01	.942± .001	.927± .001	.957± .001	.518± .006	.57± .05	.5± .1

Table 1: Performance Comparison of Different Methods. To standard deviation was estimated from training five seeds for each method, the mean performance was calculated by taking the mean prediction across the five seeds.

Reviewer Point P 3.13 — Model scores

The authors prioritize recall over precision motivated by a problem that they describe as a "needle-in-the-haystack problem".

- For a stringent argumentation that does not compare apples with pears, could streamlining the type of scores in figures, legends, text help? sklearn https://scikit-learn.org/stable/modules/generated/sklearn.metrics.classification_report.html is a machine learning library with great user acceptance and for binary classification, it suggests, e.g., recall, with class specifications sensitivity and specificity, and balanced accuracy.
- The model is not balanced - precision is weak and sensitivity is stronger, s. Fig. 6B. Can the authors comment on hyper parameter optimization with an objective function that balances recall and precision, e.g., f-score?

Reply: Thank you for suggesting these metrics. We agree that consistent metrics are important for rigorous evaluation. We highlighted enrichment curves in the main text because they best represent our model's intended use: ranking racemate-resolving agent pairs for efficient early recognition of promising candidates.

To address your point about a more comprehensive evaluation, we have added Section S5 in the Supplementary Information. There, we evaluate our model and baselines using the metrics you suggested. Our two-step training approach still shows the best performance across these metrics, reinforcing our findings.

Regarding the hyper parameter optimisation, we initially optimized hyper parameters to minimize validation BCE loss, as detailed in Section S3.4. We agree that for our use case - where missing a positive has a much greater cost than getting something that does not work - optimizing to maximize recall would likely be better. We will consider this valuable insight for future model development.

Reviewer Point P 3.14 — Parts of the approach cannot be recreated by the readers because of the lack of information.

- For recreation, the auxiliary model (auto-encoder) is needed as much as the main model (transformer encoder). Since the authors state that the training data of the auxiliary model will not be made available, the training of this model cannot be recreated. An alternative would be to publish the auto-encoder once trained, e.g., the checkpoint and an inference script to call it. Has this option been discussed with the authors?
- Details how the fingerprints were calculated, e.g., radius, size, counts or bits, are missing.

Reply: Thank you for raising this issue, we now include the checkpoint file of the auto-encoder as well. Details of Morgan fingerprints are now detailed in Section S5.2.

Reviewer Point P 3.15 — It is not obvious where to find the data. `Data/solvent_standartisation_dict.json` seems to contain only the solvent-related data. The data could be in the repo that is given in the README file. However, when I tried to clone it, I got error 404. It would be good to know whether the other reviewers can recreate this as it could be a proxy error on my side.

Reply: We apologise for not making the link to the data more prominent in the initial review. We now include a Data Availability statement in the main text which directly links to the data.

The code repository is currently private, and will be made public upon publication.

Reviewer Point P 3.16 — Statistics, Fig. 4

Can the authors comment on the lack of uncertainty bands in Fig. 4 A, B? I was under the impression that Fig. 4A, B shows the results from a cross-validation ensemble of 5 members, while Fig. 4C those of a bootstrap ensemble of 10 members.

Reply: Thank you for pointing out the lack of uncertainty bands in Figure 4A and B, and for your question about the cross-validation.

You are correct that Figures 4A and B show results from a 5-fold scaffold-split cross-validation. To clarify the process: we divided our data into five scaffold-based folds, ensuring each unique racemate was in only one fold. Then, we trained five separate models. Each model was trained on four folds and evaluated on the held-out fifth fold. Figure 4A and B were generated by combining the predictions from each of these five models on their respective held-out folds.

Therefore, while we used five models in a cross-validation scheme, each data point in Figure 4A and B is actually predicted by only *one* model, the one trained without that specific fold. This is why uncertainty bands were not initially included; it wasn't a traditional ensemble prediction where multiple models predict each point.

We understand the value of showing uncertainty. To address your comment, we have now repeated the entire 5-fold cross-validation procedure multiple times. By doing this, we now have multiple predictions for each data point, allowing us to calculate and include uncertainty bands in the updated Figure 4A and B.

Minor

Reviewer Point P 3.17 — The authors talk about mirror images and enantiomers. It depends on whether the mirror images are super-imposable. If they are not, it is an enantiomer.

Reply: We acknowledge that we are discussing mirror images of chiral molecules where the mirror images are non-superimposable, thus enantiomers.

Reviewer Point P 3.18 — Column manufacturer in Supplementary Material, Table 3. The authors list "Diacel". "Daicel" is a known manufacturer.

Reply: Fixed.

Reviewer Point P 3.19 — Missing explanations of N, K, L etc. in the legend of Fig. 1 in the Supplementary Material.

Reply: Fixed.

Response to the reviewers

We thank all reviewers for their time and constructive feedback, which has contributed to improving our manuscript. We are pleased that Reviewer 1 and Reviewer 2 now support the publication of our work.

We also thank Reviewer 3 for their continued engagement. In response to their further points, we have:

- Revised a section title to more accurately reflect its scope.
- Provided detailed clarifications on our comparison with 3DMolCSP and our model’s hyper-parameters.
- Incorporated an additional baseline (MAPC fingerprint) into our extended comparison in the Supporting Information.
- Addressed questions regarding dataset coverage and terminology.

We provide detailed responses to each of Reviewer 3’s points below.

Reviewer 1

I thank the authors for their comprehensive responses to the reviewers comments. My concerns regarding data availability and compound characterisation have been addressed and I have no further questions for the authors. I support the current version of the manuscript and SI for publication.

Reply: Thank you for your time reviewing our manuscript and for your comments, which helped us to clarify the characterisation of our experimental data. We greatly appreciate your support for its publication.

Reviewer 2

The authors have sufficiently addressed the comments made by the reviewers. In my view, the article is now suitable for publication.

Reply: Thank you for the insightful discussions during the peer review process. We believe that addressing your questions has helped us to make the nuances of diastereomeric resolution clearer. We greatly appreciate your support for publication.

Reviewer 3

The authors have taken considerable efforts which has improved the methodology in the machine learning part of the manuscript. Below only the open points in ascending priority. I suggest a second revision.

Reviewer Point P 3.1 — S4.1 and 4.1 claim ”Model accurately predicts chiral separation across chemical space” ChEMBL is the reference for drug-like small molecules. It has a broader scope,

e.g., it includes > 5x as many structures as GEOM-Drug and > 120 x as many structures as the authors use from GEOM-Drug to provide evidence for their claim.

See again the reference I sent in the first review

Chiral Cliffs: Investigating the Influence of Chirality on Binding Affinity

<https://chemistry-europe.onlinelibrary.wiley.com/doi/10.1002/cmdc.201700798>

There, it takes > 800 compounds to cover the chemical space sufficiently. UMAP is used.

Can the authors comment on why, here, only 450 compounds are sufficient evidence for their claim? I acknowledge the constraint on diastereomeric resolution but such a high discrepancy is unexpected.

Reply:

1. **Scope of the Claim** We agree that our original claim, "Model accurately predicts chiral separation across chemical space," was overly broad. We have revised the title of Section 4.1 to "Model accurately predicts chiral separation for drug-like molecules".

To better support this claim, we have included the full GEOM-Drug dataset in our analysis in Section S4.1. We note, however, that the box plots have not changed noticeably because of this. This is expected due to the law of large numbers and is the reason why we originally chose a 20k subset rather than the full dataset.

In addition, we have expanded the PCA analysis to include the ChEMBL database. Figure S4 (reproduced below) shows that our 450 training compounds (coloured points) effectively sample the high-density regions of both GEOM-Drug and the much larger ChEMBL database. The distributions of GEOM-Drug and ChEMBL are highly similar in this PCA space, confirming that GEOM-Drug is a suitable proxy for drug-like space and that our data lies within it. We do not include ChEMBL in the box-plot analysis, as calculating those statistics for >2M compounds would be computationally intensive and, given the similarity shown by the PCA, we believe it would offer limited additional insight.

2. **"Chiral Cliffs" reference** We appreciate you suggesting the "Chiral Cliffs: Investigating the Influence of Chirality on Binding Affinity" paper. We have read the paper carefully. However, we were unable to find any mention of UMAP within the main text or the supplementary information. Furthermore, the paper does not appear to explicitly discuss the concept of 'coverage of chemical space' or how many compounds might be 'sufficient' for such coverage.
3. **Sufficiency of 450 compounds** We believe that the diversity analysis we have done in Section S4 now sufficiently supports the adjusted claim that the "Model accurately predicts chiral separation for drug-like molecules".

Reviewer Point P 3.2 — 4A: The uncertainties for 3DMolCSP are so high that it happens to drop even below the most naive baseline. What could be the reasons?

Reply: As shown in Figure 4A, 3DMolCSP outperforms the two simple baselines across most of the depicted range. Furthermore, the extended comparison in Supplementary Section S5.3 demonstrates that 3DMolCSP surpasses these baselines on aggregated metrics as well.

One potential reason for the higher variance observed in 3DMolCSP is its reliance on the neural network's encoder to learn features salient for successful resolution. Given that our dataset comprises

Figure R1: Principal Component Analysis (PCA) of reference datasets using Morgan fingerprints. **A.** PCA projection of GEOM-Drug compounds. **B.** PCA projection of ChEMBL compounds. Coloured points represent our training data projected onto the same PCA space. Both reference datasets show similar distributions, with ChEMBL exhibiting greater extent due to its larger size.

6,000 resolution experiments, a scale considered modest in machine learning, as noted in your previous review, a relatively high seed variation is not surprising. This is particularly relevant in the scaffold-split setting presented in Figure 4A. For further discussion, please refer to our response to Point P 3.4.

Reviewer Point P 3.3 — l. 177: Another difference is the model inputs. I was under the impression that the authors input x, y, z conformer coordinates and one-hot encoding into 3DMolCSP, as opposed to MD results and COSMOtherm descriptors into the proposed model. Coordinates are a different abstraction level than atom densities, while the models are asked to learn the same output.

Reply:

You are correct that 3DMolCSP ingests 3D conformer coordinates and one-hot encoding of the solvent system, whereas our model uses atom-densities combined with COSMOtherm descriptors. These represent distinct abstraction levels, however, 3DMolCSP was specifically designed for coordinate-based convolutions. As the 3DMolCSP authors themselves acknowledge, their original architecture struggles to distinguish enantiomers without a dedicated “3DMolConv 2.0” enhancement to encode bond directions explicitly:

“However, we acknowledge that even though in theory this operation is sensitive to enantiomers (as demonstrated in Section S1), in practice, the model encounters the challenge of effectively learning sufficient chiral information to distinguish enantiomers. To address this challenge, we developed an enhanced elemental convolution (denoted as 3DMolConv 2.0) in an attempt to more explicitly model the bond directions by concatenating them into neighbors’ features.”

Extracting bond directions from the input is a core requirement of “3DMolConv 2.0”, which is a central component of 3DMolCSP. Adapting the 3DMolCSP model to our atom-density representation would thus require a substantial redesign, which lies beyond the scope of the present work.

Reviewer Point P 3.4 — Can the authors comment on how the complexity of the machine learning task depends on the abstraction level of the inputs?

Reply: The question regarding input abstraction level and ML task complexity touches on a significant area of machine learning research, for which we refer to seminal literature such as Bengio et al. [1]. In the context of our work and the comparison with the 3DMolCSP baseline, we offer the following

1. **Impact of Our Representation Design on Learning** Physically-informed representations, like the atom densities we utilize, can simplify the learning task by providing a more direct link between input features and the property being modelled. Our work builds on this by employing a pair-representation specifically designed for the diastereomeric acid-base pairs, emphasizing their intermolecular interactions. We believe that is one of the key contributions of our work. To further clarify this design rationale in the manuscript, we have revised the end of the second paragraph in Section 3 (Chirality-sensitive machine learning) to state:

... Here, we introduce dedicated long-range channels to capture intermolecular interactions in the acid-base pair, in addition to each atom's local environment. The final representation for a single atom consists of the mean and variance of its environment across the MD trajectory snapshots. This representation design emphasizes the intermolecular interactions within the acid-base pair, aiming to provide a more direct and physically-informed signal for downstream models.

2. **Comparison with 3DMolCSP** When comparing to 3DMolCSP, we already explicitly mention and focus on the representational advantage discussed above. See the main text (Section 4.1):

We hypothesize that our pair-representation is a key factor in our model's superior performance compared to 3DMolCSP. Our representation can directly capture differences in the interactions of enantiomer pairs with the resolving agent, while 3DMolCSP must infer these effects from individual structures. Further details on 3DMolCSP modifications and model comparisons are available in SI Section S5.

And again the the supplementary information (Section S5.3):

We hypothesize that the observed performance difference stems from representational disparities. The two neural network architectures likely have similar expressivity as indicated by the parameter counts – ours (8.5 M) and 3DMolCSP (9.3 M). However, our pair representation is potentially more informative. It can directly capture interactions between the racemate and resolving agent, whereas 3DMolCSP would need to infer these interactions indirectly.

As you have pointed out in Point P 3.2 3DMolCSP has a high seed variation. We expect that again this can be attributed to a less well-suited representation, further exacerbated by a lack of a large-scale dataset that would enable effective representation learning.

We believe our manuscript presents a fair comparison of the two approaches and that the difference in performance, rooted in these representational choices, is discussed carefully and in sufficient detail. Note, we explain why simply plugging in our representation into 3DMolCSP is out of scope for this work in our answer to Point P 3.3.

[1] Bengio, Y., Courville, A. & Vincent, P. Representation Learning: A Review and New Perspectives. IEEE Transactions on Pattern Analysis and Machine Intelligence 35, 1798–1828 (2013).

Reviewer Point P 3.5 — Table S2, S3.4: The authors explain that the choice of 19 cross-attention heads is a result of black-box optimization. With 19 heads claimed as "optimal", it needs further explanations, in my view. 19 heads means 19 queries in the attention block, implying there are 19 distinct patterns that a) can be identified from only 450 compounds and b) are necessary to learn chiral separation from the data. What are the mechanistical reasons (as opposed to treating the problem as a black box) for this high number and how much worse are the scores of models with fewer heads?

Reply: This point raises questions about our choice of model hyperparameters, the scale of our dataset, and the scope for mechanistic investigations. We address these aspects individually below:

1. **Choice of Hyperparameters:** The selection of 19 cross-attention heads was indeed determined empirically through a systematic hyperparameter optimization process (Optuna, detailed in SI Section S3.4). During this process we evaluated multiple configurations, including those with fewer heads, and found the combination used throughout the remaining of the work to be the most performant. We do not claim this specific configuration is the global optimum; however, relying on empirical performance is a standard and principled methodology in machine learning for selecting such architectural parameters.
2. **Dataset Scale:** The concern that 19 distinct patterns are being identified from "only 450 compounds" appears to conflate the number of unique chiral compounds with the actual scale of the training data. It is crucial to reiterate that the model is trained on over **6,000 distinct resolution experiments**. Each experiment is a unique combination of specific acid, base and solvent system – diastereomeric salt resolutions are not possible without having all three components. It is this full set of >6,000 experiments that provides the informational basis for the model. As we noted previously, this is analogous to natural language processing, where models learn from a vast corpus of sentences (our experiments), not just the unique words in the vocabulary (our unique compounds).
3. **Mechanistic considerations:** While one could theoretically propose that the chemical complexity of these systems (involving diverse functional groups like alcohols, amines, carboxylic acids, aromatic systems; varied steric environments; and different solvent effects) might necessitate numerous attention heads to capture a wide array of interaction types, assigning a specific chemical motif or precise interaction type to each of the 19 individual heads post-hoc would be speculative. Such detailed mechanistic deconstruction of the attention heads is not the primary objective of this work. We also note that we already include an analysis of some of the transformer layers in the model in Section 4.2. As such, we do not believe additional attention analyses would significantly strengthen our work.

We have also adjusted Section S3.4 of the SI to avoid the possibly confusing claim of optimality, and to discuss the high number of cross-attention heads:

Hyperparameters were optimised using a Tree-structured Parzen Estimator strategy as implemented in Optuna [optuna*2019]. For each trial, we sampled a configuration from a

predefined search space encompassing learning rate, batch size, network architecture parameters (depth, hidden dimensions, attention heads), dropout probability, and activation function. Each configuration was evaluated by training a model and measuring binary cross-entropy loss on a dedicated validation set. Optuna’s pruning mechanism was employed to accelerate the search by terminating unpromising trials early. The configuration yielding the lowest validation loss was subsequently used for our models presented in this work.

The optimization process identified a notably high number of cross-attention heads (19). While this could indicate the model’s need to capture diverse interaction patterns between the acid-base pair and the solvent, it is more likely a consequence of the optimization balancing data fitting against the significant regularization from dropout (\$p = 0.32\$ ).

Reviewer Point P 3.6 — Since the authors use chiral fingerprints, there may be better options than ECFP resulting in stronger benchmarks. <https://www.x-mol.net/paper/article/1790421545364107264>

Reply: We have included the MAPC fingerprint into our extended comparison in Section S5.3. Our results indicate that the Random Forest (RF) models achieved very similar performance levels when using either MAPC or Morgan fingerprints. Given this similarity, and considering that Morgan fingerprints serve as a widely recognized baseline in cheminformatics, we have retained the Morgan-based results in the main text figures to maintain clarity and avoid visual clutter.

We believe these findings highlight that, for this particular task, the choice of the learning architecture (RF vs. our NN approach) has a more significant impact on performance than the specific fingerprint used within the RF. We discuss this explicitly in the SI:

Notably, all Random Forest (RF) baseline models demonstrated similar performance, regardless of the molecular encoding employed. This held true whether using random numbers, Morgan fingerprints, or MAPC – a fingerprint specifically developed to capture chiral features. This suggests that RFs may be less suited for this specific task, as their performance lags behind neural network-based approaches, even when provided with a chirally-informed representation. Crucially, RFs are not directly compatible with the two-step training scheme that provides a significant benefit to our neural network approach.

Reviewer Point P 3.7 — I am not sure whether ”prospective” and ”retrospective” borrowed from clinical studies are the best categories here as molecules are graph data, not longitudinal data.

Reply: The terms ”retrospective validation” and ”prospective validation” are standard and accepted conventions for describing distinct approaches to model evaluation within medicinal chemistry, cheminformatics, and the broader field of applied machine learning in the sciences.

- **Retrospective validation** involves testing the model on historical data that was available prior to or during the model’s development but was strictly held out from the training process. It assesses the model’s ability to generalize to unseen examples from the existing body of knowledge. However, this type of evaluation often has indirect forms of data leakage in the form of evaluating the model on the test set multiple times during the design phase, etc.

- **Prospective validation**, on the other hand, involves making predictions with the finalized model and then experimentally testing them. These experimental data are generated **after** the model is finalised. This approach most closely mimics how the model would be used in practice.

Our use of this terminology is consistent with common practice and, we believe, accurately conveys the nature of our validation experiments. For a deeper discussion of model validation strategies within medicinal chemistry see [2].

[2] Kearnes, S. Pursuing a Prospective Perspective. Trends in Chemistry 3, 77–79 (2021).

Reviewer Point P 3.8 — Table S3: What does "To standard deviation" mean?

Reply: This was a typo. Fixed to "The standard deviation..." .

(Remarks on code availability): I can confirm that I have access to the data https://datadryad.org/share/SaY3Rs3klHnuKe1M47SW_vHpN9g7h4MkCXQ-anSG_wQ

Response to the reviewers

All reviewers now support the publication of our work. We thank all of them for their time and constructive feedback.

Reviewer 3

The authors have sufficiently addressed the points made by the reviewers. In my view, the manuscript is now suitable for publication. For the sake of completeness, here the correct reference <https://www.sciencedirect.com/science/article/abs/pii/S0021967321002181>, Fig. 6. Apologies for sending an earlier paper of the same authors, <https://chemistry-europe.onlinelibrary.wiley.com/doi/10.1002/cmdc.201700798>.

Reply: Thank you for the detailed feedback throughout the review process. We believe that while addressing your concerns we have made the manuscript significantly more rigorous.